# Coordination cages integrated into swelling poly(ionic liquid)s for guest encapsulation and separation

Xiang Zhang[1], Dawei Zhang [1] ✉, Chenyang Wei[1], Dehua Wang [2] ✉, Roy Lavendomme [3,4], Shuo Qi[5], Yu Zhu[5], Jingshun Zhang[1], Yongya Zhang [1,6], Jiachen Wang[7], Lin Xu[1], En-Qing Gao [1], Wei Yu [5], Hai-Bo Yang [1] ✉ & Mingyuan He [1] ✉

Coordination cages have been widely reported to bind a variety of guests, which are useful for chemical separation. Although the use of cages in the solid state benefits the recycling, the flexibility, dynamicity, and metal-ligand bond reversibility of solid-state cages are poor, preventing efficient guest encapsulation. Here we report a type of coordination cage-integrated solid materials that can be swelled into gel in water. The material is prepared through incorporation of an anionic $Fe^{II}_4L_6$ cage as the counterion of a cationic poly(ionic liquid) (MOC@PIL). The immobilized cages within MOC@PILs have been found to greatly affect the swelling ability of MOC@PILs and thus the mechanical properties. Importantly, upon swelling, the uptake of water provides an ideal microenvironment within the gels for the immobilized cages to dynamically move and flex that leads to excellent solution-level guest binding performances. This concept has enabled the use of MOC@PILs as efficient adsorbents for the removal of pollutants from water and for the purification of toluene and cyclohexane. Importantly, MOC@PILs can be regenerated through a deswelling strategy along with the recycling of the extracted guests.

Coordination cages, which are also known as metal-organic cages (MOCs), are a class of discrete metallosupramolecular capsules[1–5]. They are assembled from metal ions and organic ligands by coordination-driven self-assembly exploiting the reversibility of the metal-ligand bond. Although coordination cages are occasionally used in the solid state[3,6], which are treated as porous crystalline materials similar to metal-organic frameworks (MOFs)[7–10], the solubility and host-guest chemistry of these discrete cages in solution are the most appealing. The well-defined cavities are capable of binding guest molecules with high affinity and selectivity, leading to abundant applications in catalysis[11–19], storage of reactive species[20], and molecular separations[21–26]. Nevertheless, the use of soluble MOCs in solution brings about the difficulty in material recovery; The host-guest chemistry of MOCs in the solid state may also be weakened or even not

[1]State Key Laboratory of Petroleum Molecular & Process Engineering, Shanghai Key Laboratory of Green Chemistry and Chemical Processes, School of Chemistry and Molecular Engineering, East China Normal University, Shanghai 200062, PR China. [2]State Key Laboratory of Petroleum Molecular and Process Engineering, SINOPEC Research Institute of Petroleum Processing, 100083 Beijing, PR China. [3]Laboratoire de Chimie Organique, Université libre de Bruxelles (ULB), Avenue F. D. Roosevelt 50, CP160/06, B-1050 Brussels, Belgium. [4]Laboratoire de Résonance Magnétique Nucléaire Haute Résolution, Université libre de Bruxelles (ULB), Avenue F. D. Roosevelt 50, CP160/08, B-1050 Brussels, Belgium. [5]Advanced Rheology Institute, Department of Polymer Science and Engineering, Frontiers Science Center for Transformative Molecules, State Key Laboratory for Metal Matrix Composite Materials, Shanghai Jiao Tong University, Shanghai 200240, PR China. [6]College of Chemistry and Chemical Engineering, Shangqiu Normal University, Shangqiu 476000, PR China. [7]Physics Department, Shanghai Key Laboratory of Magnetic Resonance, School of Physics and Materials Science, East China Normal University, Shanghai 200062, PR China. ✉e-mail: dwzhang@chem.ecnu.edu.cn; wangdh.ripp@sinopec.com; hbyang@chem.ecnu.edu.cn; hemingyuan@126.com

survive due to the limited flexibility and dynamicity, as discussed below.

When designing new capsular hosts for specific guests, it is common to analyze the match in shape and size between them, and the 55% packing coefficient rule (i.e. the ideal filling of a host cavity by a guest bound through weak interactions) established by Rebek is useful in predicting molecular binding[27]. Apart from the shape and size complementarity, the structural flexibility and dynamic character of assembled hosts play important roles and sometimes are even crucial factors in determining the performance of guest binding[28,29]. Structural flexibility allows hosts to expand and adapt their capsular portals and internal cavities to enhance guest binding performances, analogous to the induced-fit behavior of substrates binding within the active sites of enzymes[30–32]. Reversible formation of assembled hosts from subunits in solution, on the other hand, allows capsules to open dynamically to accommodate guests facilely. As proposed by Raymond[33,34], MOCs could allow guests that are too large to fit through the windows to enter the cavity by expansion of the windows or by rupture of a metal-ligand bond. Pioneering studies from Rebek and coworkers have also demonstrated that for the dimeric tennis balls[35], softballs[36], and a cylindrical capsule[37], guest exchange occurs through openings formed by partial disruption of the hydrogen-bonding seams. Such guest binding/exchange mechanisms have also been broadly observed for other soluble organic and metal-organic capsules[38–47].

The unique positioning of gel materials at the boundary between liquids and solids offers attractive features when combining with MOCs[48]. Particular attention has been devoted to the coupling of polymers with MOCs to prepare MOC-branched (star) or crosslinked networks[49,50]. The MOC-integrated gels have shown tailored mechanical properties, self-healing ability, switchable network topology, and even permanent porosity[51–54]. We envisioned that the confined solvent within gels could provide ideal microenvironments for the coordination cages to dynamically move and flex, enabling these cages to bind guests efficiently. However, the host-guest chemistry of these integrated MOCs is rarely explored[51]. The strategy to prepare MOC-integrated gels is also limited to the use of MOCs as covalent junctions.

Poly(ionic liquid)s (PILs), frequently consisting of polycationic/anionic frameworks with counterions, have the ability to swell into gels upon deliberate design[55,56]. Considering the ionic nature of both coordination cages and PILs, we propose in this work a versatile strategy to prepare a type of swellable MOC@PIL gels through incorporation of anionic $Fe^{II}_4L_6$ cages as the counterions of cationic imidazolium-based PILs (Fig. 1). The integration of MOCs has been found to greatly affect the swelling ability, mechanical properties, and morphology of the gels. Importantly, the swelling of MOC@PILs retains the flexibility and dynamic character of the soluble metallohost in water for the binding of various guests, including those that are larger than the size of the cage portals. This concept enables the use of MOC@PILs as efficient and regenerable adsorbents for the removal of pollutants from water and for the purification of organic chemicals.

## Results

### Design, synthesis, and characterization

Based upon the ionic nature of both coordination cages and swellable PILs, an ion exchange strategy was designed to combine the two. We infer the swelling and volume expansion of a polycationic framework of PIL allow molecules of a soluble anionic cage to sufficiently diffuse into the porous gel and exchange with the counterions of PILs. For such purpose, an anionic $Fe^{II}_4L_6$ cage (Fig. 2a, tetramethylammonium (TMA$^+$) as the counterion), which was reported and widely studied by the Nitschke group[20,57–60], was selected as the metallohost. This cage is water-soluble and highly stable, and has shown fruitful guest binding properties in water. Following the reference[57], the synthesis is presented in Supplementary Fig. 1. On the other hand, an imidazolium-based PIL (Fig. 2a, $NO_3^-$ as the counterion) was deliberately designed to maximize the swellability in water and was synthesized through radical copolymerization (Supplementary Figs. 4–6). Imidazolium polymeric chains, associated with ethylene glycol crosslinkers and hydrophilic counterions such as $NO_3^-$, were reported to have high swelling ability in water[56], and a swelling capacity of 350 was obtained for our prepared PIL-$NO_3^-$. Acetone was proven to be a favorable solvent to extract the absorbed water from the swollen PIL, resulting in the recovery of PIL-$NO_3^-$ to its initial non-swollen agglomerate state (Fig. 2b).

Through simple stirring of the synthesized PIL with the $Fe^{II}_4L_6$ cage in water, ion exchange took place, resulting in obvious color

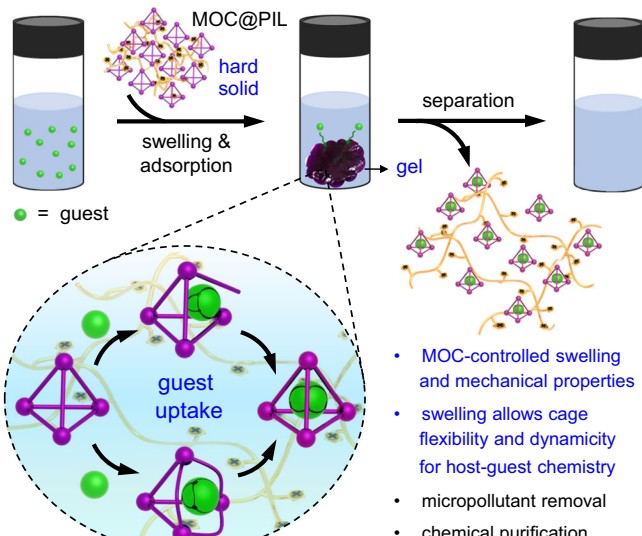

**Fig. 1 | Schematic illustration of the concept of this work.** The enlarged inset highlights the two possible binding mechanisms, i.e. portal expansion and vertex dissociation, of a metal-organic cage for the guest molecule that is larger than the portal but can fit within the cavity.

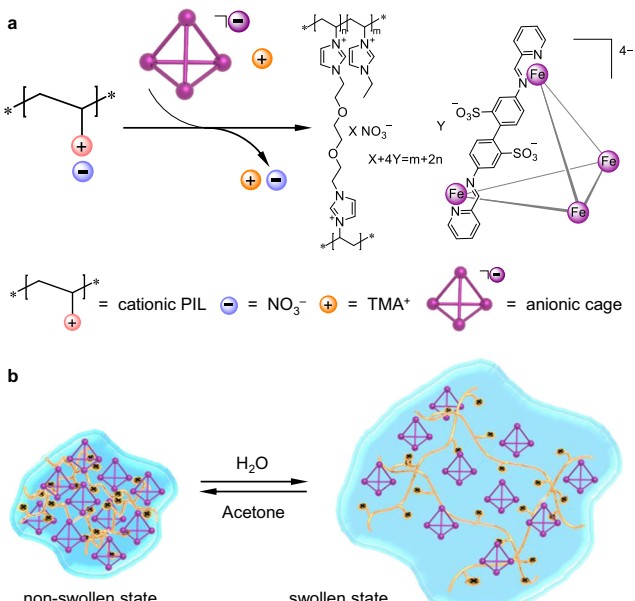

**Fig. 2 | Synthesis and swellable behavior of MOC@PILs. a** Illustration of the synthesis of MOC@PILs through an ion exchange strategy. **b** Illustration of the swelling and deswelling behavior of MOC@PILs by adding suitable solvents.

change of the material from colorlessness to purple (Fig. 2a). The precipitate was obtained by centrifugation and was washed with pure water repeatedly until no color resulting from the free $Fe^{II}_4L_6$ cage was observed in the supernatant. The addition of a large amount of acetone enabled the cage-gel composite to agglomerate. The amount of the $Fe^{II}_4L_6$ cage in the composite could be quantified by ICP-AES for the measurement of Fe after digesting the sample with nitric acid. We infer the amount of the detected Fe to be exclusively from the intact immobilized MOCs due to the high stability of the $Fe^{II}_4L_6$ cage based on the following two sets of experiments: (i) $^1H$ NMR experiments indicated the integrality of the $Fe^{II}_4L_6$ cage in $D_2O$ without any decomposition in the presence of a large excess of PIL monomers (12 equiv.) (Supplementary Fig. 10); (ii) After saturating PIL-$NO_3^-$ with an excess of cage for immobilization, only pure cage could be observed in the solution with no trace of decomposition species, including any subcomponents (Supplementary Fig. 12). Samples (including MOC@PILs **1**–**6**) with varying loadings of the $Fe^{II}_4L_6$ cage from 0.11 to 0.74 g/g were thus prepared.

A direct evidence for the successful immobilization of the cage onto the PIL chains was provided by a second ion exchange. The addition of an excess of sodium nitrate into the solution containing swelling MOC@PILs could release the anionic $Fe^{II}_4L_6$ cage into the solution to pair with $Na^+$. The $^1H$ NMR spectrum recorded for this solution proved integrality of the cage, the proton signals of which were fully consistent with the dissolved cage (Supplementary Fig. 13). In particular, the singlet of $TMA^+$, which was the initial counterion of the cage, disappeared on the spectrum, confirming the occurrence of ion exchange for the immobilization.

A series of other characterization methods were also performed. (i) The FTIR spectra of MOC@PILs presented peaks from both the $Fe^{II}_4L_6$ cage and the imidazolium polymer (Supplementary Fig. 14). In particular, upon increasing the content of the cage within MOC@PILs from **1** to **6**, the characteristic peaks of the cage in the region of 1285–965 $cm^{-1}$ continuously increased, with gradual weakening of the specific peak of $NO_3^-$ (1385 $cm^{-1}$). This observation indicated the occurance of anion exchange between the anionic cages and the nitrates. (ii) The solid-state $^{13}C$ MAS NMR spectrum of MOC@PIL **6**

showed a series of typical signals resulting from both the anionic cage and the cationic polymer (Supplementary Fig. 15). (iii) Thermogravimetric measurements indicated decomposition of the PIL and the cage beginning at 205 and 300 °C, respectively, while intermediate temperatures were displayed for thermal degradation of MOC@PILs (Supplementary Fig. 16). Moreover, the fraction of metal oxide residues was observed to increase from samples **1** to **6**.

## Swellability, mechanical properties, and mophology of MOC@PILs

Agglomerate samples of MOC@PILs could rapidly swell in water and reached equilibrium within 20 min (Supplementary Fig. 18), resulting in a series of purple hydrogels. Importantly, the swelling capacity of MOC@PILs could be adjusted by altering the loadings of the immobilized cages. When the cage loading was lower than 0.32 g/g, the swelling capacity of MOC@PILs in water was not significantly altered. For instance, the swelling capacity of MOC@PIL **1** (Q = 317, cage loading, 0.32 g/g) was very close to that of the parent PIL-$NO_3^-$ (Q = 350). Interestingly, upon reaching this threshold of cage loading (0.32 g/g), the values of Q were observed to decrease almost linearly from MOC@PIL **1** to **6** (Fig. 3a, c). A maximum cage loading of 0.74 g/g for MOC@PIL **6** led to a swelling capacity of only 5. We infer this phenomenon to have two main causes: 1) the anionic cages are more hydrophobic than nitrate anions, which reduces the affinity of the composite with water; 2) each cage with four negative charges requires to be surrounded by four imidazolium cations for charge balance, serving as a noncovalent crosslinker to more densely link the cationic PIL chains to prevent their swelling in water.

To demonstrate the electrostatic interactions between the immobilized anionic cages and the cationic PIL chains, $^1H$ NMR spectra for the fully swollen gels of PIL-$NO_3^-$ and MOC@PILs **1**–**6** were recorded. Results showed that with the increase of the cage loading from gels **1** to **6**, gradual upfield shifts of proton signals of the imidazolium and ethyl groups were observed, consistent with the expected interactions (Supplementary Fig. 19). We also measured the $^1H$ NMR spectra of MOC@PIL **1** at different swelling degrees (50, 100, 200, 317-fold) by adding differing amounts of water. Unnoticeable changes between

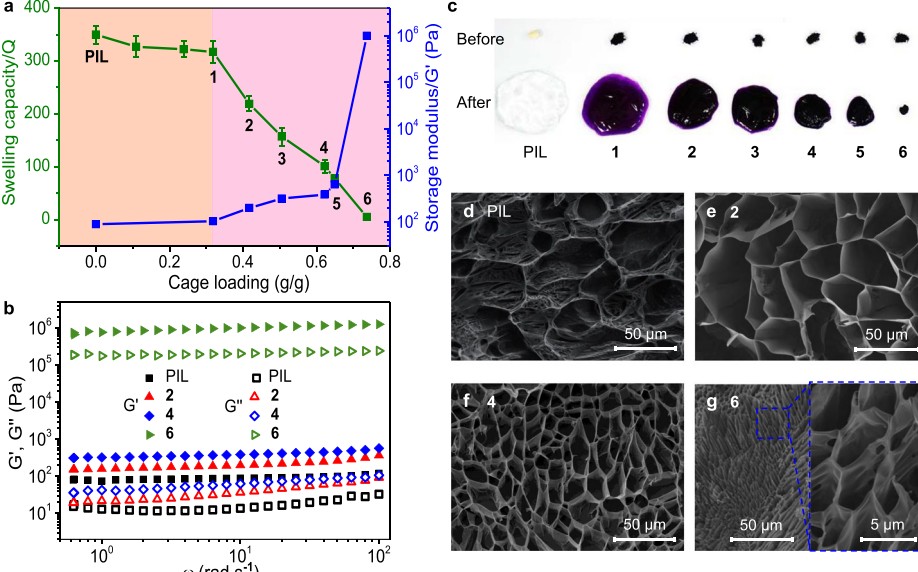

**Fig. 3 | Swelling capacity, mechanical properties, and morphology of swollen MOC@PILs. a** Swelling capacity and storage moduli G′ of MOC@PILs **1**–**6** and the parent PIL-$NO_3^-$. Error ranges were calculated from triplicate experiments. **b** Storage and loss moduli, G′ and G″, of the swollen PIL-$NO_3^-$ and MOC@PILs **2**, **4**, and **6** as a function of angular frequency ω. **c** Photographs of PIL-$NO_3^-$ and MOC@PILs **1**–**6** before and after swelling. **d**–**g** Cryo-SEM images of swollen PIL-$NO_3^-$ and MOC@PILs **2**, **4** and **6** after freeze-drying showing the microstructures.

spectra of **1** were observed, indicating the swelling degree was unable to perturb the strength of the interactions between the immobilized MOCs and the PIL chains (Supplementary Fig. 20).

The decrease of swelling capacity and enhancement of gel stiffness with increasing MOC loadings were confirmed through rheological studies. Frequency sweep tests at 0.1% shear strain showed the storage moduli (G') dominated over the loss moduli (G'') for all the swollen gels within the tested range, indicating strong elastic response from all the samples (Fig. 3b and Supplementary Fig. 21). The G' value of hydrogel **1** was 103 Pa at an oscillatory frequency of 10.05 rad s$^{-1}$, close to the value of the parent PIL-NO$_3^-$ (G' = 89 Pa) (Fig. 3a). In contrast, the gradual increase of the cage loading (>0.32 g/g) within MOC@PILs witnessed concomitant increase of G', suggesting the increased stiffness. A G' of $1.01 \times 10^6$ Pa was obtained for MOC@PIL **6** with a maximum cage loading of 0.74 g/g, corresponding to a 10$^4$-fold increase with respect to the parent PIL-NO$_3^-$. These rheology results agree with the tests of swelling capacity discussed above, revealing the role of anionic crosslinkers played by the immobilized cages[54]. Moreover, time-dependent oscillatory tests demonstrated high stability of all these swollen gels (Supplementary Fig. 22).

The Cryo-SEM images of swollen MOC@PILs after freeze-drying revealed interconnected porous network of the gels (Fig. 3d–g), in contrast to the initial agglomerate state prior to swelling (Supplementary Fig. 23). Akin to the microstructure of the parent PIL-NO$_3^-$, expanded pores are present in a regular layout within the swollen MOC@PILs, while the diameters of which continuously decreased from **2** (30–50 μm) to **4** (10–30 μm) and **6** (3–6 μm). This result was consistent with the observation of macroscopic volume contraction from **2** to **6** (Fig. 3c), and demonstrated the significant effect of cage loading on the morphology and pore size of swollen MOC@PILs. The presence of the large honeycomb-like pores, even for the swollen MOC@PIL **6** having the highest cage loading, is beneficial for mass transfer and cage accessibility, ensuring sufficient contact of the anionic cages with substrates for host-guest interactions.

## Guest binding properties of MOC@PILs

The Fe$^{II}_4$L$_6$ cage was previously reported by the Nitschke group to bind a set of hydrophobic guests in water, such as benzene, fluorobenzene, cyclohexane, cyclohexene, dioxane, CH$_2$Cl$_2$, and CHCl$_3$ (Fig. 4a)[57–60]. The sizes of these guest molecules are larger than those of the cage portals observed from the crystal structure (Supplementary Fig. 25). To further reveal the importance of cage flexibility for guest inclusion, we explored four additional guests, norbornane, norbornene, norbornadiene, and 7-oxabicycloheptane (Fig. 4a). These new bicyclic guests were found to bind within the Fe$^{II}_4$L$_6$ cage in a 1:1 stoichiometry and slow exchange dynamics on the NMR time scale were observed (Supplementary Figs. 27–38). Binding costants were determined and could be found in Supplementary Table 2.

Note that the four new bicyclic guests have an extended three-dimensional shape and are larger in both diameter and volume than those previous guests (Supplementary Table 1). In particular, the V/Ω$_A$ (V for molecular volume; Ω$_A$ for asphericity) parameters of the bicyclic guests, which are correlated to the binding kinetics[60], are one order of magnitude larger than those of the monocyclic guests (Supplementary Table 1). Molecular modeling of the host-guest complexes suggested the expansion of the cage cavity (Supplementary Fig. 26). The necessity of heating to facilitate binding equilibration further indicated that mechanisms of both portal expansion and vertex dissociation (Fig. 1) may be at work on account of the high energy required[33,34,40]. Moreover, we have also performed binding experiments of the cage in the solid state by adding cage solids directly into the liquids of pure guest (norbornadiene and 7-oxabicycloheptane). Results proved failure of guest uptake into the cage cavities (Supplementary Fig. 39), also highlighting the importance of the flexibility of dissolved cages in water.

With the solution binding performance of the Fe$^{II}_4$L$_6$ cage in hand, we explored guest binding behavior of the immobilized cage within MOC@PILs. The broad proton signals of MOC@PILs after swelling prevented us to investigate guest binding through $^1$H NMR, however, simple $^{19}$F NMR spectra provided a direct evidence for guest uptake with the hydrogel material. As shown in Fig. 4c, when an excess of fluorobenzene was added into an aqueous solution containing swollen MOC@PIL **1**, a new resonance at −106.5 ppm was observed, along with the peak of free fluorobenzene (−113.8 ppm). The new signal was attributed to the encapsulated guest[61], indicating the encapsulation of fluorobenzene within MOC@PIL **1**. We could also use the approach of ion exchange for investigation of binding the guests having no fluorine atoms. The addition of an excess of nitrate to (guest⊂MOC)@PILs released the anionic species guest⊂MOC from gels into solution (Fig. 4b). The $^1$H NMR spectra of the resulting solution presented intact host-guest complexes (Fig. 4d and Supplementary Figs. 41–51). To our delight, all of the guests listed in Fig. 4a, that can be bound by the soluble cage, could be bound by MOC@PILs as well, demonstrating the retaining of the guest binding properties after cage immobilization.

To evaluate the influence of swelling of MOC@PILs on guest binding kinetics, we first measured the kinetic curves of the soluble cage and the immobilized cage within MOC@PIL **6** for binding three guests, benzene, cyclohexane, and norbornane. The experiments were conducted through mixing the host, either the soluble or the immobilized cage, with guest-saturated aqueous solution, and the percentage of the occupied host was monitored by $^1$H NMR spectroscopy. Results showed that for the smallest benzene, the binding kinetic curves of the soluble cage and the immobilized cage (MOC@PIL **6**) almost overlapped, while binding of cyclohexane and norbornane with MOC@PIL **6** were much slower than with the soluble cage (Supplementary Fig. 52). Moreover, we also monitored the concentration decrease of the free guests (initially fixed at 10 ppm) when in the presence of swollen MOC@PILs (**2, 4, 6**; 1.5 equiv. of the immobilized cage relative to the guest). MOC@PILs with higher swellability were found to generally present faster guest removal (Supplementary Fig. 53). These results reveal the effect of swelling on guest uptake kinetics: The higher degree of swelling allows immobilized cage to be more flexible and dynamic to present faster binding kinetics, while small guests, such as benzene, that are relatively easy to enter the cavity, could be bound rapidly even using MOC@PIL **6** that has the lowest swellability.

We have also determined the apparent binding constants of MOC@PILs for the guests to investigate the effect of swelling on binding thermodynamics (Supplementary Tables 3 and 4). As shown in Fig. 5a, when we used MOC@PIL **6** (Q = 5) that had the lowest swellability as the host, the apparent K$_a$ for various guests were generally smaller than those with the soluble cage. Moreover, the reduction in binding affinity became increasingly significant upon increasing the size of guests. We have also measured the apparent K$_a$ with swollen MOC@PILs **1**–**6** having different levels of swelling for benzene and norbornane (Supplementary Table 4). Results showed that for smaller benzene, no obvious alteration in apparent K$_a$ with swollen MOC@PILs across all levels of swellability was observed, and the values were very close to the value with the soluble cage (Fig. 5b). In contrast, the apparent K$_a$ for larger norbornane progressively increased upon increasing the swellability of MOC@PILs. These results demosntrate that the swellability is also able to modify the binding thermodynamics of the immobilized cages within MOC@PILs. We infer that the alteration of the binding affinity for the immobilized cage results from the distinct microenvironments: The immobilized cage is surrounded by cationic PIL chains and may become constrained relative to the free cage, while this effect is alleviated if the constraint around the cage loosens.

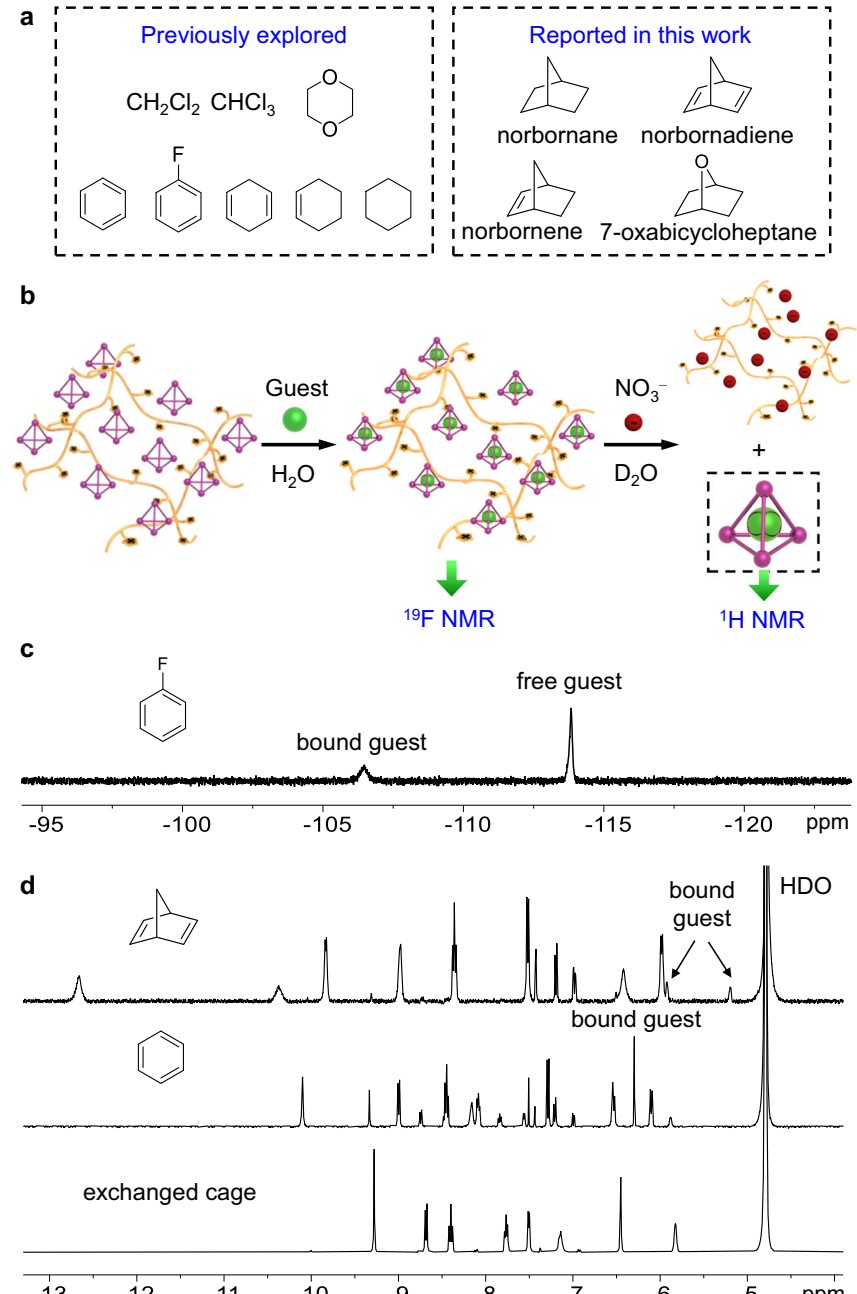

**Fig. 4 | Host-guest chemistry of MOC@PILs. a** Guests investigated in this work, including those that were previously reported by the Nitschke group[57–60] and the four new bicyclic guests. **b** Schematic illustration of the strategy for investigation of the host-guest chemistry of MOC@PILs. **c** ${}^{19}$F NMR (D$_2$O, 470 MHz, 298 K) spectrum of MOC@PIL **1** in the presence of an excess of fluorobenzene. **d** ${}^{1}$H NMR (D$_2$O, 400 MHz, 298 K) spectra of the anionic cage, benzene⊂MOC, and norbornadiene⊂MOC released from MOC@PIL **6** or (guest⊂MOC)@PIL **6** by adding an excess of NaNO$_3$.

## Pollutant removal and chemical purification

Pollutant treatment is critical in modern society and adsorption is a leading technology for removing pollutants from water[62,63]. The above results suggested the great potential of using swollen MOC@PILs as efficient and regenerable adsorbents for water purification. In this context, we chose MOC@PIL **6** as the adsorbent due to the high cage loading and appropriate stiffness after swelling, which could be easily recycled after adsorption through centrifugation or filtration. The low swellability of **6** (Q = 5) also consumed only a tiny amount of water for in-situ swelling, allowing a large proportion of water sample to be left. The above adsorption kinetic experiments of **6** (Supplementary Fig 53) also indicated the capability of complete removal of benzene, cyclohexane, and norbornane within a reasonable period of time.

As shown in Fig. 6, agglomerate MOC@PIL **6** swelled in polluted water containing pollutants (the molar ratio between immobilized cage and pollutant was 1.2 or higher), and guest binding occurred. After thorough mixing and equilibration, the swollen (pollutant⊂MOC)@PIL **6** was separated from the purified water via centrifugation. ${}^{1}$H NMR spectra of the purified D$_2$O indicated that pollutants CH$_2$Cl$_2$, CHCl$_3$, benzene, and 1,4-cyclohexadiene were removed efficiently (≥94%) after 2 h at room temperature (rt) (Supplementary Figs. 54–57). Besides, almost complete removal of cyclohexene, cyclohexane, norbornadiene, norbornene, and norbornane from water were also exhibited after 6 h at 50 °C (Supplementary Figs. 59–63). The harsher extraction conditions of the latter resulted from the relatively larger size of these guests that had slower binding

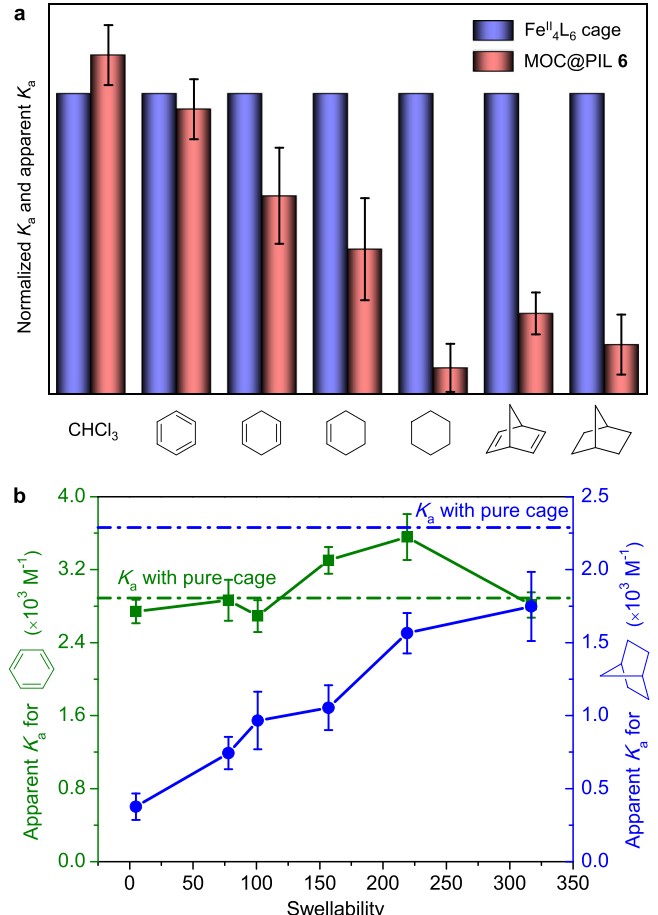

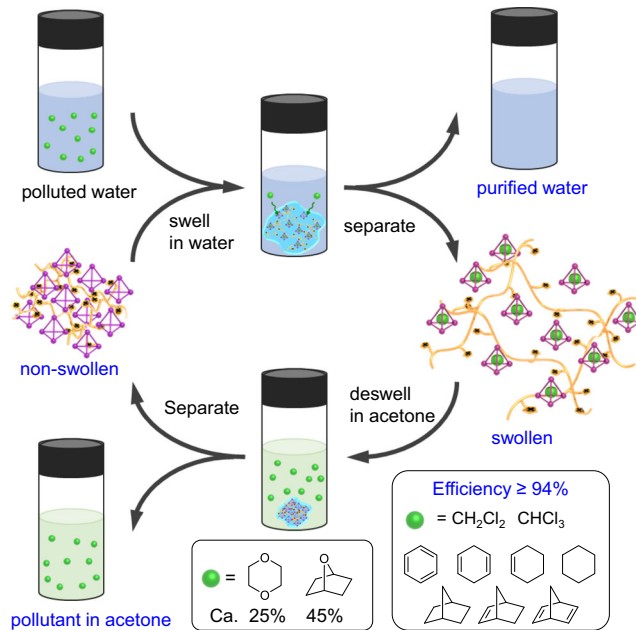

**Fig. 6 | Schematic illustration of water purification using MOC@PIL 6.** For the removal of guests $CH_2Cl_2$, $CHCl_3$, benzene, 1,4-cyclohexadiene, cyclohexene, cyclohexane, norbornadiene, norbornene, and norbornane from water, efficiency higher than 94% could be achived. For guests 1,4-dioxane and 7-oxabicycloheptane, removal efficiencies of 25% and 45% were respectively obtained.

**Fig. 5 | Guest binding affinities of MOC@PILs. a** Normalized $K_a$ of the $Fe^{II}_4L_6$ cage and the apparent $K_a$ of MOC@PIL **6** for a series of guests with differing sizes in water. **b** Apparent $K_a$ of MOC@PILs **1**–**6** for benzene and norbornane in water. The dashed green and blue lines represent the binding constants of the soluble cage for benzene and norbornane, respectively. Error ranges were calculated from triplicate experiments.

kinetics. 1,4-Dioxane and 7-oxabicycloheptane, the two polar organic pollutants, could be also captured, although their efficiencies were lower than others (25% and 45% for 1,4-dioxane and 7-oxabicyclo-heptane, respectively) (Supplementary Figs. 58 and 64).

Note that pure PIL-$PF_6^-$ (Q = 12) was unable to get rid of these pollutants from water, suggesting the important role of the immobilized cage. PIL-$PF_6^-$, instead of PIL-$NO_3^-$, was used as the control adsorbent due to the appropriate swellability of the former: the swellability of PIL-$NO_3^-$ (Q = 350) was too high and no supernatant water remained for analysis after in-situ swelling of the adsorbent. Importantly, the extracted guests and MOC@PIL **6** could be separately recycled. The addition of acetone to the swollen (guest⊂MOC)@PIL **6** could extract the absorbed water within the gel together with the bound guest. After stirring for 30 min, MOC@PIL **6** deswelled and was isolated by centrifugation; the guest was found to completely transfer to the acetone phase (Supplementary Figs. 66 and 67). Through this strategy, sample MOC@PIL **6** could be reused at least for five times with no significant decrease in efficiency (Supplementary Fig. 68).

We then set about investigating the use of MOC@PILs to build systems capable of chemical purification[64–70]. Considering the $Fe^{II}_4L_6$ cage could bind benzene but not toluene, MOC@PIL **6** was used in the purification of toluene by removing a trace amount of benzene. As shown in Fig. 7, after mixing toluene/benzene and the fully swollen MOC@PIL **6** at rt, the trace amounts of contained benzene could be

extracted into the swollen **6**. The purity of toluene (wt%) was found to be increased from 97.9% to 99.2% detected by GC (Supplementary Fig. 69). Moreover, as the binding kinetics of benzene and 1,4-cyclo-hexadiene were much faster than that of cyclohexane, we took advantage of this phenomenon to purify cyclohexane that simultaneously contained trace amounts of benzene and cyclohexadiene. Results showed that a purity of 98.1% for a cyclohexane sample (containing 0.9% benzene and 1.0% cyclohexadiene) was raised to 99.2% (containing 0.1% benzene and 0.7% cyclohexadiene) (Supplementary Fig. 70). We could use the same deswelling method, as described in water treatment, to regenerate the adsorbents. These results demonstrate the potential of swollen MOC@PILs as a promising type of supramolecular adsorbents to design systems for chemical purification.

## Discussion

We have proposed the potential of using swelling of polymers to control the flexibility, dynamicity, and even guest binding mechanisms of coordination cages. This concept was demonstrated by integration of anionic cages into swellable cationic PILs through ion exchange, resulting in a series of MOC@PILs having differing cage loadings. The amount of the immobilized cage within MOC@PILs were found to control the swelling ability and mechanical properties of MOC@PILs. In comparison to the conventional modulation of MOC junctions to tune the mechanical properties of MOCs-branched or crosslinked gels[48,49], the ion exchange strategy developed here was simpler and more straightforward. Importantly, the swelling of MOC@PILs modified both the kinetics and thermodynamics of guest uptake, and higher degrees of swelling enabled the immobilized cages to bind guests similarly to the soluble cages. The swollen MOC@PILs were developed as a promising type of supramolecular adsorbents, which were efficient in removal of organic pollutants from water and in purification of organic chemicals. Moreover, through the strategy of deswelling, the MOC@PIL adsorbents could be regenerated.

It should be emphasized that many coordination cages, including both anionic and cationic types, with varying guest binding properties

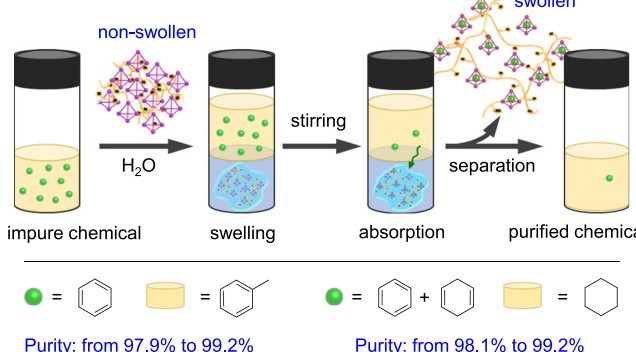

**Fig. 7 | Schematic illustration of chemical purification using MOC@PIL 6.** The purity of toluene was increased from 97.9% to 99.2% after removal of benzene by selective adsorption. The purity of cyclohexane was raised from 98.1% to 99.2% after selective adsorption of benzene and cyclohexadiene.

have been reported and can be combined with either cationic or anionic PIL chains. Our strategy is versatile and allows the preparation of customized MOC@PILs with desired cage loadings, swelling abilities, and mechanical properties, for separation of particular targets. The efficiency, stability, and reusability suggest great potential of this type of cage-integrated material for molecular separation towards practical application. The concept developed in this work may also be extended to the development of regenerable cage-based soft material for supramolecular catalysis.

## Methods

4,4′-Diaminobiphenyl-2,2′-disulfonic acid (75%, containing water) and tetramethylammonium hydroxide pentahydrate (98%) are ordered from Energy Chemical. 2-Formylpyridine (98%) is ordered from Adamas-beta. FeSO$_4$·7H$_2$O (99%) is ordered from Greagent. AgNO$_3$ (99.8%) is ordered from Sinopharm Chemical Reagent. 1-Vinyl-3-ethylimidazolium bromide (VEIMBr) (99%) is ordered from Innochem Co., Ltd.

NMR spectra were recorded using a Bruker 400 MHz Avance III HD Smart Probe ($^1$H, $^{13}$C and $^{19}$F NMR and 2D experiments). Chemical shifts for $^1$H, $^{13}$C and $^{19}$F NMR are reported in ppm on the $\delta$ scale; $^1$H and $^{13}$C were referenced to the residual solvent peak. Coupling constants ($J$) are reported in Hz. Fourier transform infrared spectra (FTIR) were recorded on a Nicolet NEXUS 670 spectrophotometer using KBr pellets in the 4000–500 cm$^{-1}$ regions. Thermogravimetric analyses (TGA) were carried out on a Mettler Toledo TGA/SDTA851 instrument at the heating rate of 10 °C min$^{-1}$ in the temperature range of 30–800 °C under flowing air. Scanning electron microscopy (SEM) was performed on a ZEISS GeminiSEM 300 microscope. Cryo-SEM was carried out on a FEI Quanta 450 including refrigerating equipment Quorum PP3000T. Fe contents were measured by inductively coupled plasma atomic emission spectrometry (ICP-AES) using an IRIS Intrepid II XPS spectrometer. All solid-state magic angle spinning NMR (ss MAS NMR) experiments were conducted on a Bruker Avance III NMR spectrometer at 600 MHz (14.1 Tesla). A standard Bruker 4 mm triple-resonance HXY MAS probe was used to collect all spectra at room temperature, with a spinning frequency of 10 kHz. Solid-state $^{13}$C MAS NMR spectra were performed with regular $^1$H decoupling pulse sequence, with a 70 w 4 µs pulse (90°) for $^{13}$C and 30k scans. The $^{13}$C chemical shifts were referenced to adamantane signal at 38.48 ppm. Rheology experiments were performed on a Kinexus rheometer. A parallel-plate with a radius of 12.5 mm was used, and coupled with a bottom plate (25 mm in diameter). The gap between the two plates was ~1 mm. Frequency sweep experiments were conducted from 0.1 to 100 rad/s at 0.1% strain, which was in the linear viscoelastic regime as

confirmed using strain sweep experiments. Time-dependent oscillatory tests were carried out at 0.1% strain (also in the linear viscoelastic regime). Gas chromatography (GC) measurements for water samples were recorded on a Shimadzu GC-2014 instrument configured with a FID detector and a DB-WAX UI column (30 m × 0.25 mm × 0.25 µm). The following GC method was used: the oven was programmed in an initial temperature at 50 °C, then ramp at 5 °C min$^{-1}$; the injection temperature was 140 °C; the detector temperature was 140 °C. The samples were injected with a volume of 2.0 µL. GC measurements for the samples generated during the experiments of chemical purification were recorded on a Shimadzu GC-2014 instrument configured with a FID detector and a HP-INNOWax column (30 m × 0.32 mm × 0.25 µm). The following GC method was used: the oven was programmed in an initial temperature at 50 °C and remained for 2 min, then ramp at 10 °C min$^{-1}$; the injection temperature was 220 °C; the detector temperature was 220 °C. The samples were injected with a volume of 0.2 µL.

## Data availability

The authors declare that the data supporting this study are available within the paper and its supplementary information file. Additional data are available from the corresponding author upon request.

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

## Acknowledgements

This work was supported by the National Natural Science Foundation of China (22201075). R.L. is a Postdoctoral Researcher of the Fonds de la Recherche Scientifique – FNRS. The authors are grateful for financial support from the SINOPEC Research Institute of Petroleum Processing and East China Normal University.

## Author contributions

D.Z., D.W., H.-B.Y., M.H. and X.Z. conceived and designed the research. X.Z. carried out the majority of the experimental work. C.W. performed preliminary experiments. R.L. carried out theoretical calculations. S.Q. performed rheology tests. J.W. performed solid-state NMR experiments. Y.Z., J.Z., Y.Y.Z., L.X., E.-Q.G. and W.Y. contributed to the analysis and interpretation of the results. X.Z. and D.Z. wrote the initial draft of the manuscript. All authors edited the manuscript.

## Competing interests

The authors declare no competing interests.
