## [Peer Review File · Nature Communications]

Coordination Cages Integrated into Swelling Poly(Ionic Liquid)s for Guest Encapsulation and SeparationREVIEWER COMMENTS

Reviewer #1 (Remarks to the Author):

In this paper, the authors describe the incorporation of Nitschke's well-known M4L6 cage into a poly-ionic liquid via electrostatic interactions, and an application of the complex towards separating hydrocarbons from aqueous solution.

There are interesting concepts in this paper – attaching coordination cages to solids is tricky, and there still aren't that many examples. This is an ingenious way to do that. Studying the effects of the polymer on the host has merits. The use of the system as an extractant is slightly interesting, and is successful. The technical aspects are broadly acceptable.

However, for publication in Nat Comm, the question of novelty arises. The cage is (very, very) well known, as are its binding abilities. The application towards separation is novel, but is never going to be useful for actual real-world separations due to cost, sensitivity of the cage, selectivity of the binding (inhibition) and scale, so it's purely a proof-of-principle. Nitschke has shown a number of extractions with water-soluble cages, some of anions, some of hydrocarbons, so the concept of "cage as extractor" isn't novel, only the PIL-cage complex is. Therefore, the impact lies in novel structure/behavior or study of molecular recognition, and as discussed below, this is lacking. I recommend publication elsewhere, probably a materials-specialist journal, after a comprehensive rewrite.

Main text:

The introduction is trying to introduce the concept of "guests larger than the portals of the cage", to link with some observations seen later. However, it's far too short and undercited to achieve this, and focuses solely on metal-organic cages, when the concept has been explored thoroughly with other hosts as well. There are many, many examples of guests binding in hosts that are larger than the portals, it's not uncommon. Many enclosed capsules exist that show this behavior that are not cited – Rebek, Reinhoudt, Gibb, Fujita, Mukherjee, Yoshizawa, Raymond, Nitschke. The observation simply isn't novel.

The second paragraph in the introduction is far too simplistic: "When designing new hosts for specific guests, it is common to analyze the match in shape and size between them. This requirement, however, assumes the host cavity to be static and rigid." This simply isn't true, there are myriad examples of molecular recognition in flexible receptors. The paragraph is designed as a segue to the author's application, but is misleading as written. This also links to the main technical problem with the paper (see below).

The terms associative and dissociative in Figure 1 are wrong. The two mechanisms are possible, they're fine, but they need to be called something else - associative and dissociative are specific mechanistic terms in inorganic (and supramolecular) chemistry, referring to the specific mechanism of a reaction. The mechanisms shown in Figure 1 are not these mechanisms, so need to be labeled differently.

The results are a little opaque – p4, para 1: “For such purpose, an anionic FeII4L6 cage (Fig. 2A, tetramethylammonium (TMA+) as the counterion), was selected as the metallohost. The synthetic procedures and characterization data are presented in Supplementary Section 2.1. This cage was water-soluble and was shown to have fruitful guest binding properties in water”. This cage has been known since 2008, and scores of papers have been published using it. To say that these authors showed its guest binding properties in water (rather than specifically mentioning the Nitschke group) is misleading. This may be a mis-phrasing, but it looks very bad as written. This also comes up again in p8 – Fig 4 shows the “previously done” vs “done here” ok, but the text does really read this way. I would also check whether Nitschke has bound those 4 guests in his cages – he has definitely encapsulated them in other M4L6 cages of equivalent size. I can't remember whether they were bound in this cage before, but it is highly likely. He has far more papers on the guest properties of this than ref 59.

The authors' argument that the PILs are somehow involved in allowing the binding of “guests larger than the portal size” is very strange. The binding properties of this cage and many, many others have been known for decades, reviewed many times, and in many cases, the guests require ligand flexing to get in. Fujita's whole point about “Molecular Panels” in 2001 was that the panels slow guest binding and enable discrete, long-lived Michaelis complexes. Rebek (not cited) published a whole series of papers on this in the late 1990s, with various mechanisms for the kinetics. More importantly, the binding properties of this cage are very, very well-known, and are identical in solution to that shown in the PIL. So, while it's interesting that the cage was put in the PILK, and the swelling, etc, is interesting, the whole discussion of molecular recognition is not representative of the literature.

The concept of volume of guest (p10) is also misleading – cyclic guests rotate to fill the space in the cavity, so can have binding affinities close to that of bicyclic “3D” guest of larger volume (again, Rebek published papers on this in the 2000s, not cited).

P12: “The higher degree of swelling allows immobilized cage to be more flexible and dynamic to present binding performances similar to the level under solution-state conditions even for larger guests” – there is zero evidence for this presented by the authors, other than the result of K_a . That's not enough to ascribe mechanistic details. It could be a kinetic phenomenon with the PIL. Also, the “portal opening” concept is KINETIC, not thermodynamic – this has effects on entry RATES, but not necessarily binding affinities, so this manner of analysis is flawed if the goal is to analyze mechanism of ingress. The authors mix rates and affinities in a very un-quantitative way.

Technical points:

The binding affinities in cage are ok (as they've been done before), but the apparent binding constants of cage-PIL are only estimates, as the authors state that not all the cage is released from the PIL. In addition, the concentration of free guests, many of which are volatile, could change upon centrifugation – this makes the accuracy of the numbers shown in S-Table 3 very weak. Certainly they should not be stated to 3 sig figs, and should only be described as estimates. The chart in the text (Fig 5) is probably ok, as all the affinities are relative and any errors should be constant. But I couldn't find out how the authors performed their error treatment, either. Overall, the S-Table affinities for cage:PIL should be far more tentative in their descriptions, and not promise accuracy that is impossible.

The 2D NMRs are ok, but messy, and should be shown as contour plots, not bitmaps. The contours cannot be seen as shown.

Reviewer #2 (Remarks to the Author):

In this article, the authors report the preparation and application of a hybrid material constructed from a poly(ionic liquid) (PIL) and a known negatively charged metal-organic cage (MOC). Different loadings of MOC to PIL were explored, and the materials and swelling were characterized using a number of techniques. The applications of the hybrid material in micropollutant removal from water were explored; additionally, the authors have reported novel host-guest chemistry for this MOC and used this information to extend the scope of the micropollutant removal studies.

I think this is a very clever approach to the construction of complex materials based on simple underlying principles. For me it is a clear step forward from existing work and is thus novel enough for publication in Nat. Comm. However, certain areas of discussion in this require further consideration before publication is recommended.

The discussion in the introduction focusses on the applications of MOCs in the solution and solid-state, but some attention should be given to existing examples that bridge this gap to provide fuller context/precedent for this work. The incorporation of MOCs into polymer hydrogels is certainly relevant here, and although several references have been included to highly relevant manuscripts in this area, I think the introduction must explicitly describe this precedent. I think that the incorporation of MOCs into porous liquids is also very relevant. A recent manuscript describing permanently porous ionic liquid gels based on metal-organic polyhedra (DOI: 10.1021/jacs.3c03778) should be acknowledged.

The authors state they used ICP-AES to determine the amount of the MOC immobilized in the PIL after digestion with nitric acid. I believe that this assay can detect the amount of Fe immobilized in each sample, not the amount of in-tact MOC (which is a difficult question to address). This quantification does not account for the possibility that some of the cage within the PIL could have decomposed before the digestion step.

The next experiment, in which the cage is released from the PIL after ion exchange and analysed by ^1H NMR goes some way to addressing this – however, it doesn't seem to be quantitative and would not be able to detect any non-coordinated Fe present. We still do not know what the loading of in-tact cage in each sample is. Later experiments (FTIR, ^{13}C MAS NMR and TGA) show increased loadings in the different samples but cannot give an absolute loading.

In the section on micropollutant removal – experiments to monitor the removal of CH_2Cl_2 , CHCl_3 , benzene, 1,4-cyclohexadiene, cyclohexane, cyclohexene, norbornadiene, norbornene and norbornane from D_2O were conducted by NMR and the authors state the removal efficiency was “close to 100%”. I would be cautious of using NMR experiments to validate such claims given the low sensitivity compared other techniques such as GC (which is used later). What's the minimum concentration of these micropollutants that can actually be detected using such experiments? The number of scans performed should be reported and the authors should acknowledge the limitations of NMR in their discussion.

Additionally, in the SI, the authors state “The concentration of the micropollutant after adsorption was analyzed by ^1H NMR spectroscopy, and tert-butanol or ethylene glycol was used as the internal standard”. However, no concentrations calculated in this way or the exact removal efficiencies are actually reported – so we cannot determine how “close to 100%” the efficiencies really are.

Reviewer #3 (Remarks to the Author):

This article describes the fabrication of new composite materials based on water-swelling poly(ionic liquid)s (PILs) and coordination cages for guest capture and separation. The concept is very interesting to use the swelling PILs to provide a solution-like microenvironment for the cages, which allows the cages to be dynamic and bind guests, followed by the deswelling process for recycling. In reading the abstract and introduction of the manuscript, I had high expectations for the work, but it turned out to be overselling and lacked mechanistic studies. Therefore, I do not recommend this current version be accepted in Nature Communications. The authors should address the following severe scientific concerns.

1. The authors attributed the change of cage@PILs swelling capacity to be the outcome of (a) cage hydrophobicity and (b) the limited motion of PIL chains due to their electrostatic interaction with cages. As far as I understand, the cages with four negative charges can serve as crosslinkers to more densely link the PIL chains to prevent their swelling in water. However, the authors did not demonstrate any experiments to support this hypothesis. For instance, the authors could have demonstrated the mesoscale characterization of composites after freeze-drying. However, the only cage@PIL 4 was characterized by SEM. The authors should check all the mesoscale structures with different cage loading and discuss how the resulting morphology, size, and width would be affected by the loading amount. The authors should show evidence to support their hypothesis.

2. Related to the above discussion, the authors should also carry out the macroscopic characterization such as gel rheology (viscoelastic property). The quantitative analysis of mechanical properties can be correlated to the interaction between PIL and the cages. The authors only mention;

“The maximum cage loading led to a swelling ratio of 5 for cage@PIL 6 (0.74 g/g), the texture of which was relatively stiff.”

The authors should quantitatively characterize this stiffness by a conventional rheological method.

3. The authors claimed on page 12 as follows;

“These results demonstrate the important effect of swelling on guest binding: The higher degree of swelling allows immobilized cage to be more flexible and dynamic to present binding performances similar to the level under solution-state conditions even for larger guests”.

I still do not understand this statement. The authors should separately consider the macroscopic swelling behavior from the microscopic crosslinking between PILs and cages. The former should influence the water uptake within the gels, which means the hydration of the cages and concentration of the cages. The latter is truly the electrostatic interaction between PILs and the cages. The authors should consider how the swelling process would influence the interaction between PILs and cages. Is it possible to determine this interaction by carefully analyzing any NMR technique?

4. Related to the above, the strange behavior was observed in Fig. 3a. This is not a linear correlation. But there is a threshold at the lower concentration domain of the cage loading. What about the concentration lower than 0.32 g/g for cage@PIL 1? It seems a certain concentration is essential for the crosslinking of PILs by the cage molecules to influence the swelling property.

5. The authors did not describe why cage@PIL 6 was chosen as the representative adsorbent for the micropollutant removal and purification. This sample possesses the lowest swelling capacity and can be easily recycled from the solution without the deswelling process. As far as I understand, the addition of acetone here is to help the release of the encapsulated organic molecules rather than the deswelling. To

study the effect of sample swelling behavior, the authors should compare the pollutant extraction performance among cage@PILs with different cage loading amounts.

6. Regarding the extraction experiment of micropollutant, the sample of cage@PIL 6 showed complete removal of quite different types of molecules. Due to its low swelling capacity, the extraction of bigger molecules was shown to take longer time and require higher temperatures. The authors should show the kinetics adsorption data as a function of time and compare it with other cage@PILs to figure out the role of the swelling process of composites.

7. More importantly, the authors demonstrated the “micropollutant” removal experiments with very high concentrations of pollutants, which is not realistic for micropollutants. Indeed, the concentration was not shown in the main text and I needed to check the experimental sections in the supporting information. According to the SI, the concentration of micropollutants is fixed at 5 mM, which might be in the range of a few hundred ppm. These values are at least 5 orders of magnitude higher than the practical situation because a typical concentration range of organic micropollutants is between a few ppb and tens of ppt. The authors should not call this experiment “micropollutant” removal.

8. The authors used PIL-PF6- as the control example to claim that the micropollutants were extracted only by cages. The authors should explain why PIL-PF6- was used instead of the PIL-NO3 initially used for the composite synthesis.

Point by point response to the reviewers:

Referee: 1

In this paper, the authors describe the incorporation of Nitschke's well-known M_4L_6 cage into a poly-ionic liquid via electrostatic interactions, and an application of the complex towards separating hydrocarbons from aqueous solution.

There are interesting concepts in this paper – attaching coordination cages to solids is tricky, and there still aren't that many examples. This is an ingenious way to do that. Studying the effects of the polymer on the host has merits. The use of the system as an extractant is slightly interesting, and is successful. The technical aspects are broadly acceptable.

However, for publication in *Nat Comm*, the question of novelty arises. The cage is (very, very) well known, as are its binding abilities. The application towards separation is novel, but is never going to be useful for actual real-world separations due to cost, sensitivity of the cage, selectivity of the binding (inhibition) and scale, so it's purely a proof-of-principle. Nitschke has shown a number of extractions with water-soluble cages, some of anions, some of hydrocarbons, so the concept of "cage as extractor" isn't novel, only the PIL-cage complex is. Therefore, the impact lies in novel structure/behavior or study of molecular recognition, and as discussed below, this is lacking. I recommend publication elsewhere, probably a materials-specialist journal, after a comprehensive rewrite.

Reply: We greatly appreciate the critical comments from the reviewer, as well as the very helpful suggestions in the report. We have carefully addressed all the suggestions from the reviewer, as well as the suggestions from Reviewers 2 and 3, as detailed below. Several new results have been added into the revised MS and substantial revisions have been made. With these changes and responses, we think the quality of the manuscript has been greatly improved.

Nevertheless, we still would like to explain a few words on the novelty of this work, which we believe was not clear enough in the original MS. We report here a new type of metal-organic-cage-integrated gel materials (MOC@PIL) through a simple strategy of ion exchange. The immobilized cages within MOC@PILs were found to greatly affect the swelling ability of the gels and the mechanical properties. The preparation of MOC-integrated gels has received a growing interest, as widely explored by Jeremiah A. Johnson (*Nature* **560**, 65-69 (2018); *Nat. Chem.* **8**, 33-41 (2016); *J. Am. Chem. Soc.* **145**, 21879-21885 (2023)). In comparison to the conventional modulation of MOC junctions to tune the mechanical properties of MOCs-branched or crosslinked gels (*Acc. Chem. Res.* **51**, 2437-2446 (2018); *Matter* **4**, 2123-2140 (2021)), the ion exchange strategy developed here is simpler and more straightforward. Importantly, unlike the normal solid-state cages that have weak flexibility and dynamic character, which are the prerequisites for guest binding, the swelling and uptake of solvent into MOC@PILs provided an ideal solution-like microenvironment within the gels for the immobilized cages to dynamically move and flex that retained the excellent solution-state guest binding performances. This concept enabled the use of MOC@PILs as efficient and recyclable adsorbents for the removal of pollutants from water and for the purification of toluene and cyclohexane.

As noted by the reviewer, the cage is very well known, as well as its binding abilities. We chose this cage to prepare the cage-integrated gels because of its high stability in water and fruitful guest binding abilities that can be conveniently utilized with our newly developed MOC@PILs. Moreover, our strategy is versatile. Many mature coordination cages, including both anionic and cationic types, with varying guest binding properties have been reported and can thus be combined with either cationic

or anionic PIL chains following this strategy. We have thoroughly checked the reports from the Nitschke group using cages as extractors and found three cases were reported. Apart from the use of soluble cages to transport cargoes across liquid membranes (*J. Am. Chem. Soc.*, **143**, 12175-12180 (2021)), cages have been used for separation of ReO_4^- (*Angew. Chem. Int. Ed.* **57**, 3717-3721 (2018)) and polyaromatic hydrocarbons (*J. Am. Chem. Soc.* **141**, 18949-18953 (2019)), the first author of the latter two is also one of the corresponding authors in this work. Note that in all these cases, the cages used are dissolved in solution, posing a practical problem in material recovery. Differently, MOC@PILs as efficient soft-solid adsorbents could be easily recycled in a heterogeneous fashion. We agree with the reviewer that the use of supramolecular cages for actual real-world separations is difficult, which is also a common challenge in this area. However, we hope the incorporation of cages into other types of materials with distinct properties could push their practicability a little bit further.

1. The introduction is trying to introduce the concept of “guests larger than the portals of the cage”, to link with some observations seen later. However, it’s far too short and undercited to achieve this, and focuses solely on metal-organic cages, when the concept has been explored thoroughly with other hosts as well. There are many, many examples of guests binding in hosts that are larger than the portals, it’s not uncommon. Many enclosed capsules exist that show this behavior that are not cited – Rebek, Reinhoudt, Gibb, Fujita, Mukherjee, Yoshizawa, Raymond, Nitschke. The observation simply isn’t novel.

Reply: As noted by the reviewer, we are trying to explain in the introduction that guests larger than the portals of cages could be bound as well due to the flexibility and dynamic character of the soluble cages. This is actually a distinct advantage of assembled hosts in solution. This point does not represent the novelty of this work, whereas the novelty resides in how to maintain the flexibility and dynamic character of the cage in the solid state to have solution-level guest binding performance. We greatly appreciate the suggestion from the reviewer, and agree that such behaviour is not solely the merit of metal-organic cages. For instance, the group of Rebek has reported many examples to deal with guest binding/exchange mechanisms with assembled organic capsules (a typical review: *Org. Biomol. Chem.* **2** (2004)). We have thus modified the introduction with the addition of discussion on examples of all assembled capsules, including both metal-organic cages and purely organic cages. The contributions from Rebek, Reinhoudt, Gibb, Fujita, Mukherjee, Yoshizawa, Raymond, Nitschke have been cited and highlighted in the revised MS.

MS, Page 3:

Most frequently, the binding and release of guest requires rupture of multiple weak assembling forces to create a suitable opening of capsules, as these capsules usually possess large cavities with small windows. As proposed by Raymond (Fig. 1),^{33,34} MOCs as receptors could allow guests that are too large to fit through the windows to enter the cavity by expansion of the windows or by rupture of a metal-ligand bond. Pioneering studies from Rebek and coworkers have also demonstrated that for the dimeric tennis balls,³⁵ softballs,³⁶ and a cylindrical capsule,³⁷ guest exchange occurs through openings formed by partial disruption of the hydrogen-bonding seams. Such guest binding/exchange mechanisms have also been broadly observed for other soluble organic and metal-organic capsules.³⁸⁻⁴⁷

MS, Page 20:

35. Szabo, T., Hilmersson, G. & Rebek, J. Dynamics of assembly and guest exchange in the tennis ball. *J. Am. Chem. Soc.* **120**, 6193-6194 (1998).

36. Santamaría, J., Martín, T., Hilmersson, G., Craig, S.L. & Rebek, J. Guest exchange in an encapsulation complex: A supramolecular substitution reaction. *Proc. Natl. Acad. Sci. U.S.A.* **96**, 8344-8347 (1999).

37. Craig, S.L., Lin, S., Chen, J. & Rebek, J. An NMR study of the rates of single-molecule exchange in a cylindrical host capsule. *J. Am. Chem. Soc.* **124**, 8780-8781 (2002).
38. Zhang, D., *et al.* Enantiopure [Cs⁺/Xe⊂cryptophane]⊂Fe^{II}₄L₄ hierarchical superstructures. *J. Am. Chem. Soc.* **141**, 8339-8345 (2019).
39. Akine, S., Miyashita, M. & Nabeshima, T. A closed metallomolecular cage that can open its aperture by disulfide exchange. *Chem. Eur. J.* **25**, 1432-1435 (2019).
40. Escobar, L., Escudero-Adan, E.C. & Ballester, P. Guest exchange mechanisms in mono-metallic Pd^{II}/Pt^{II} -cages based on a tetra-pyridyl calix[4]pyrrole ligand. *Angew. Chem. Int. Ed.* **58**, 16105-16109 (2019).
41. Fujita, M., *et al.* Molecular paneling via coordination. *Chem. Commun.*, 509-518 (2001).
42. Yazaki, K., *et al.* Polyaromatic molecular peanuts. *Nat. Commun.* **8**(2017).
43. Banerjee, R., Chakraborty, D. & Mukherjee, P.S. Molecular barrels as potential hosts: from synthesis to applications. *J. Am. Chem. Soc.* **145**, 7692-7711 (2023).
44. Hof, F., Nuckolls, C., Craig, S.L., Martín, T. & Rebek, J. Emergent conformational preferences of a self-assembling small molecule: Structure and dynamics in a tetrameric capsule. *J. Am. Chem. Soc.* **122**, 10991-10996 (2000).
45. Yamanaka, M., Shivanyuk, A. & Rebek, J. Kinetics and thermodynamics of hexameric capsule formation. *J. Am. Chem. Soc.* **126**, 2939-2943 (2004).
46. Avram, L., Wishard, A.D., Gibb, B.C. & Bar - Shir, A. Quantifying guest exchange in supramolecular systems. *Angew. Chem. Int. Ed.* **56**, 15314-15318 (2017).
47. Prins, L.J., Verhage, J.J., de Jong, F., Timmerman, P. & Reinhoudt, D.N. Enantioselective noncovalent synthesis of hydrogen-bonded double-rosette assemblies. *Chem. Eur. J.* **8**(2002).

2. The second paragraph in the introduction is far too simplistic: “When designing new hosts for specific guests, it is common to analyze the match in shape and size between them. This requirement, however, assumes the host cavity to be static and rigid.” This simply isn’t true, there are myriad examples of molecular recognition in flexible receptors. The paragraph is designed as a segue to the author’s application, but is misleading as written. This also links to the main technical problem with the paper (see below).

Reply: We appreciate the suggestion from the reviewer and agree that our initial statement is too simplistic and is not precise. We have thus corrected the Introduction to highlight the importance of flexibility in molecular recognition in the revised MS.

MS, Page 2:

Coordination cages, which are also known as metal-organic cages (MOCs), are a class of discrete metallocapsules.¹⁻⁵ They are assembled from metal ions and organic ligands by coordination-driven self-assembly exploiting the reversibility of the metal-ligand bond. Although coordination cages are occasionally used in the solid state,^{3,6} which are treated as porous crystalline materials similar to metal-organic frameworks (MOFs),⁷⁻¹⁰ the solubility and host-guest chemistry of these discrete cages in solution are the most appealing. The well-defined cavities are capable of binding guest molecules with high affinity and selectivity, leading to abundant applications in catalysis,¹¹⁻¹⁹ storage of reactive species,²⁰ and molecular separations.²¹⁻²⁶ Nevertheless, the use of soluble MOCs in solution brings about the difficulty in material recovery; The host-guest chemistry of

MOCs in the solid state may also be weakened or even not survive due to the limited flexibility and dynamicity, as discussed below.

Guest-binding is a complex process and usually gives rise to the most thermodynamically stable host-guest complexes. When designing new capsular hosts for specific guests, it is common to analyze the match in shape and size between them, and the 55% packing coefficient rule (*i.e.* the ideal filling of a host cavity by a guest bound through weak interactions) established by Rebek is useful in predicting molecular binding.²⁷ Apart from the shape and size complementarity, the structural flexibility and dynamic character of assembled hosts play important roles and sometimes are even crucial factors in determining the performance of guest binding.^{28,29}

Structural flexibility allows hosts to expand and adapt their capsular portals and internal cavities to enhance guest binding performances, analogous to the induced-fit behavior of substrates binding within the active sites of enzymes.³⁰⁻³² Reversible formation of assembled hosts from subunits in solution, on the other hand, allows capsules to open dynamically to accommodate guests facilely.

MS, Page 20:

27. Mecozzi, S. & Rebek, J.J. The 55% solution: a formula for molecular recognition in the liquid state. *Chem. Eur. J.* **4**, 1016-1022 (1998).

28. Palmer, L.C. & Rebek, J.J. The ins and outs of molecular encapsulation. *Org. Biomol. Chem.* **2**(2004).

29. Martin Diaz, A.E. & Lewis, J.E.M. Structural flexibility in metal-organic cages. *Front. Chem.* **9**, 706462 (2021).

30. Boehr, D.D., Nussinov, R. & Wright, P.E. The role of dynamic conformational ensembles in biomolecular recognition. *Nat. Chem. Biol.* **5**, 789-796 (2009).

31. Ronson, T.K., League, A.B., Gagliardi, L., Cramer, C.J. & Nitschke, J.R. Pyrene-edged Fe^{II}₄L₆ cages adaptively reconfigure during guest binding. *J. Am. Chem. Soc.* **136**, 15615-15624 (2014).

32. Zhang, T., Zhou, L.P., Guo, X.Q., Cai, L.X. & Sun, Q.F. Adaptive self-assembly and induced-fit transformations of anion-binding metal-organic macrocycles. *Nat. Commun.* **8**, 15898 (2017).

3. The terms associative and dissociative in Figure 1 are wrong. The two mechanisms are possible, they're fine, but they need to be called something else - associative and dissociative are specific mechanistic terms in inorganic (and supramolecular) chemistry, referring to the specific mechanism of a reaction. The mechanisms shown in Figure 1 are not these mechanisms, so need to be labelled differently.

Reply: Many thanks to the reviewer for pointing this out. Although the two terms are inherited from the original references (*Chem. Soc. Rev.* **36**, 161-171 (2007); *J. Am. Chem. Soc.* **127**, 7912-7919 (2005)), we agree with the reviewer that the two terms are misleading and should be named differently. Following the suggestion from the reviewer, “associative” and “dissociative” have been changed to “portal expansion” and “vertex dissociation” to describe more precisely the two guest binding mechanisms in the revised MS.

MS, Page 3, Fig. 1:

Fig. 1 | Schematic illustration of two guest binding mechanisms. Possible binding mechanisms of a metal-organic cage for the guest molecule that is larger than the portal but can fit within the cavity.

MS, Page 11:

The necessity of heating to facilitate binding equilibration further indicated that mechanisms of both portal expansion and vertex dissociation (Fig. 1) may be at work on account of the high energy required.^{33,34,40}

- The results are a little opaque – p4, para 1: “For such purpose, an anionic $\text{Fe}^{\text{II}}\text{L}_6$ cage (Fig. 2A, tetramethylammonium (TMA^+) as the counterion), was selected as the metallohost. The synthetic procedures and characterization data are presented in Supplementary Section 2.1. This cage was water-soluble and was shown to have fruitful guest binding properties in water”. This cage has been known since 2008, and scores of papers have been published using it. To say that these authors showed its guest binding properties in water (rather than specifically mentioning the Nitschke group) is misleading. This may be a mis-phrasing, but it looks very bad as written. This also comes up again in p8 – Fig 4 shows the “previously done” vs “done here” ok, but the text does really read this way. I would also check whether Nitschke has bound those 4 guests in his cages – he has definitely encapsulated them in other M_4L_6 cages of equivalent size. I can’t remember whether they were bound in this cage before, but it is highly likely. He has far more papers on the guest properties of this than ref 59.

Reply: We are sorry for not having noticed the misleading way we presented and we apologise for our initial phrasing. We have modified the corresponding sentences and also the caption of Figure 4 in the revised MS to highlight the significant contribution of the Nitschke group. Moreover, we have thoroughly checked the published literatures on this cage, and as noticed by the reviewer, there are 12 papers in total. We note that five of the papers have reported guest binding properties of the cage (*Science* **324**, 1697-1699 (2009); *Angew. Chem. Int. Ed.* **47**, 8297-8301 (2008); *Chem. Commun.* **47**, 457-459 (2011); *Chem. Eur. J.* **19**, 3374-3382 (2013); *J. Am. Chem. Soc.* **135**, 7039-7046 (2013)), all of which have been added into the citation of the revised MS. In particular, none of these papers have described the four new guests, norbornane, norbornene, norbornadiene, and 7-oxabicycloheptane, and we are also completely sure that they have never been involved in any literatures from the Nitschke group after carefully checking. Although exploration of the four new guests for the cage is not the main focus of this work, these guests represent the largest sizes and volumes of guests included in this specific $\text{Fe}^{\text{II}}\text{L}_6$ cage reported to date. The binding of such large guests with MOC@PILs can highlight the high flexibility of the immobilized cage within the gels and thus the importance of swelling for guest binding.

MS, Page 4:

For such purpose, an anionic $\text{Fe}^{\text{II}}\text{L}_6$ cage (Fig. 2a, tetramethylammonium (TMA^+) as the counterion), which was reported and widely studied by the Nitschke group,^{20,57-60} was

selected as the metallohost. This cage is water-soluble and highly stable, and has shown fruitful guest binding properties in water. Following the reference,⁵⁷ the synthesis and characterization data are presented in Supplementary Section 2.1.

MS, Page 10:

The Fe^{II}₄L₆ cage was previously reported by the Nitschke group to bind a set of hydrophobic guests in water, such as benzene, fluorobenzene, cyclohexane, cyclohexene, dioxane, CH₂Cl₂, and CHCl₃ (Fig. 4a).⁵⁷⁻⁶⁰

MS, Page 10:

Fig. 4 | Host-guest chemistry of MOC@PILs. (a) Guests investigated in this work, including those that were previously reported by the Nitschke group⁵⁷⁻⁶⁰ and the four new bicyclic guests. (b) Schematic illustration of the strategy for investigation of the host-guest chemistry of MOC@PILs.

MS, Page 19:

20. Mal, P., Breiner, B., Rissanen, K. & Nitschke, J.R. White phosphorus is air-stable within a self-assembled tetrahedral capsule. *Science* **324**, 1697-1699 (2009).

57. Mal, P., Schultz, D., Beyeh, K., Rissanen, K. & Nitschke, J.R. An unlockable-relockable iron cage by subcomponent self-assembly. *Angew. Chem. Int. Ed.* **47**, 8297-8301 (2008).

58. Riddell, I.A., Smulders, M.M., Clegg, J.K. & Nitschke, J.R. Encapsulation, storage and controlled release of sulfur hexafluoride from a metal-organic capsule. *Chem. Commun.* **47**, 457-459 (2011).

59. Ronson, T.K., *et al.* Size-selective encapsulation of hydrophobic guests by self-assembled M₄L₆ cobalt and nickel cages. *Chem. Eur. J.* **19**, 3374-3382 (2013).

60. Smulders, M.M., Zarra, S. & Nitschke, J.R. Quantitative understanding of guest binding enables the design of complex host-guest behavior. *J. Am. Chem. Soc.* **135**, 7039-7046 (2013).

5. The authors' argument that the PILs are somehow involved in allowing the binding of "guests larger than the portal size" is very strange. The binding properties of this cage and many, many others have been known for decades, reviewed many times, and in many cases, the guests require ligand flexing to get in. Fujita's whole point about "Molecular Panels" in 2001 was that the panels slow guest binding and enable discrete, long-lived Michaelis complexes. Rebek (not cited) published a whole series of papers on this in the late 1990s, with various mechanisms for the kinetics. More importantly, the binding properties of this cage are very, very well-known, and are identical in solution to that shown in the PIL. So, while it's interesting that the cage was put in the PIL, and the swelling, etc, is interesting, the whole discussion of molecular recognition is not representative of the literature.

Reply: We appreciate the helpful comment from the reviewer and feel sorry for not having explained clearly the novelty of this work in the original MS. We are actually not trying to demonstrate that "binding of guests larger than the portal size is strange". Instead, how to maintain flexibility and dynamic character of the cage in the solid state to have solution-level guest binding performance is difficult and unusual. Here we use the concept of swelling degree of polymers to control the flexibility and dynamic character of the integrated cages, allowing excellent solid-state guest binding properties of MOC@PIL identical to that in solution. Importantly, as mentioned by the reviewer, the groups of Rebek and Fujita have provided many evidences to demonstrate the requirement of ligand flexing for guest binding (two typical reviews respectively from Rebek and Fujita: *Org. Biomol. Chem.* **2**(2004); *Chem. Commun.*, 509-518 (2001)). These can be supportive knowledge and excellent background, the discussion of which have been added into the revised Introduction and the revised References (please see the answer for Q1).

6. The concept of volume of guest (p_{10}) is also misleading – cyclic guests rotate to fill the space in the cavity, so can have binding affinities close to that of bicyclic “3D” guest of larger volume (again, Rebek published papers on this in the 2000s, not cited).

Reply: Many thanks to the reviewer for pointing this out. Although comparison of binding affinities of cyclic and bicyclic “3D” guests is out of the focus of the manuscript, we fully agree with the reviewer that cyclic guests could rotate to fill the cavity space to have binding affinities close to that of bicyclic “3D” guests. Indeed, for both types of guests, ^1H NMR spectra of host-guest complexes present only one set of signals for both occupied cages and included guests (Supplementary Figs. 35-45), indicating the rapid rotation of guests within the cavities. We infer that the shape of guests may not significantly affect the thermodynamics (binding affinities), while the inclusion kinetics might be altered as we discussed in the revised Supporting Information (Supplementary section 4.1).

Moreover, we would feel very thankful to the reviewer if we could get some information on the specific paper the reviewer mentioned. We have carefully checked all the papers of Rebek published during 1995-2009, while failed in knowing it. Some of the papers have shown encapsulation of bicyclic guests (such as *Science* **281**, 1842-1845 (1998)), but without any comparison of binding constants with those of monocyclic guests. In the paper proposing the 55% rule (*Chem. Eur. J.* **4**, 1016-1022 (1998)), binding constants of both mono- and bicyclic guests are determined, which are shown similar, but the reason of rotation of guests within the cavity is not mentioned. Some other papers are focused on the motion and dynamics of mono/bicyclic guests (*J. Am. Chem. Soc.* **119**, 11701-11702 (1997); *Org. Lett.* **10**, 5397-5400 (2008); *Eur. J. Org. Chem* 2722-2728 (2007)), but binding constants are not compared one another. There is a very nice review from Rebek talking about molecular behavior in confined spaces (*Acc. Chem. Res.* **42**, 1660-1668 (2009)), but the fact of guest rotation is not discussed either. Nevertheless, if we combine investigations and conclusions of different papers, a comprehensive analysis may prove the statement. We also worry that we cannot carefully read all the papers and such a statement might be neglected due to our carelessness.

Supplementary Information, Page S14:

All guests studied herein are significantly larger than the portal of the static $\text{Fe}^{\text{II}}\text{L}_6$ cage from single crystal data (1.7 Å diameter for CCDC 784594, see Supplementary Figure 20). In a dynamic system, the conformational motion of the cage is expected to enlarge its entrance portals thus allowing some small guests (e.g. CH_2Cl_2) to enter the cavity. Such a conformation motion is, however, insufficient to permit the entrance of larger guests. Based on the models, the shortest dimensions of bicyclic guests (5.45–5.97 Å) are longer than those for monocyclic guests (3.54–5.27 Å), while both are significantly larger than the diameter of the cage portal (1.7 Å). Moreover, it was previously shown that the volume and shape, described by their asphericity Ω_A (see Section 8), of guests can be correlated to their inclusion kinetics following the value V/Ω_A (inverse correlation).^[5] As listed in Supplementary Table 1, the V/Ω_A parameter of new bicyclic guests studied herein (norbornane derivatives) is one order of magnitude larger than monocyclic cyclohexane derivatives, which is consistent with the much slower inclusion kinetics observed for bicyclic guests. Moreover, the above analysis doesn't take into consideration the conformational flexibility of guests. For instance, the transition between chair and boat conformations of cyclohexane or 1,4-dioxane could facilitate their insertion in the cage portals. The new bicyclic guests (norbornane derivatives) are larger and conformationally more rigid than monocyclic cyclohexane derivatives which further exacerbates the difference in expected inclusion kinetics.

Note that although the volumes of monocyclic guests are smaller than those of bicyclic guests that may affect the inclusion kinetics, monocyclic guests could rotate to fill the cavity space to have binding affinities close to that of bicyclic “3D” guests.

7. P12: “The higher degree of swelling allows immobilized cage to be more flexible and dynamic to present binding performances similar to the level under solution-state conditions even for larger guests” – there is zero evidence for this presented by the authors, other than the result of K_a . That’s not enough to ascribe mechanistic details. It could be a kinetic phenomenon with the PIL. Also, the “portal opening” concept is KINETIC, not thermodynamic – this has effects on entry RATES, but not necessarily binding affinities, so this manner of analysis is flawed if the goal is to analyze mechanism of ingress. The authors mix rates and affinities in a very un-quantitative way.

Reply: After hearing the comment from the reviewer, we acknowledge the problem of our initial statement. We agree with the reviewer that the results of K_a should represent only the thermodynamics of binding, while the effect of swelling on “portal opening” of the immobilized cages should be relevant to the inclusion kinetics. Following the suggestion from the reviewer, we have supplied additional experiments on binding kinetics. These results have been added into the revised MS and the revised Supplementary Information, as detailed below. The effect of swelling on binding kinetics and thermodynamics have been also separately discussed in the revised MS.

MS, Page 12:

To evaluate the influence of swelling of MOC@PILs on guest binding kinetics, we first measured the kinetic curves of the soluble cage and the immobilized cage within MOC@PIL **6** for binding three guests, benzene, cyclohexane, and norbornane (for details of the measurement, see Supplementary Section 5.2). The experiments were conducted through mixing the host, either the soluble or the immobilized cage, with guest-saturated aqueous solution, and the percentage of the occupied host was monitored by ^1H NMR spectroscopy. Results showed that for the smallest benzene, the binding kinetic curves of the soluble cage and the immobilized cage (MOC@PIL **6**) almost overlapped, while binding of cyclohexane and norbornane with MOC@PIL **6** were much slower than with the soluble cage (Supplementary Fig 46). Moreover, we also monitored the concentration decrease of the free guests (initially fixed at 10 ppm) when in the presence of swollen MOC@PILs (**2**, **4**, **6**; 1.5 equiv. of the immobilized cage relative to the guest). MOC@PILs with higher swellability were found to generally present faster guest removal (Supplementary Fig 47). These results reveal the effect of swelling on guest uptake kinetics: The higher degree of swelling allows immobilized cage to be more flexible and dynamic to present faster binding kinetics, while small guests, such as benzene, that are relatively easy to enter the cavity, could be bound rapidly even using MOC@PIL **6** that has the lowest swellability.

We have also determined the apparent binding constants of MOC@PILs for the guests to investigate the effect of swelling on binding thermodynamics (for details of measurement, see Supplementary Section 5.3). As shown in Fig. 5a, when we used MOC@PIL **6** ($Q = 5$) that had the lowest swellability as the host, the apparent K_a for various guests were generally smaller than those with the soluble cage. Moreover, the reduction in binding affinity became increasingly significant upon increasing the size of guests. We have also measured the apparent K_a with swollen MOC@PILs **1-6** having different levels of swelling for benzene and norbornane (Supplementary Table 4). Results showed that for smaller benzene, no obvious alteration in apparent K_a with swollen MOC@PILs across all levels of swellability was observed, and the values were very close to the value with the soluble cage (Fig. 5b). In contrast, the apparent K_a for larger norbornane progressively increased upon increasing the swellability of MOC@PIL. These results demonstrate that the swellability is also able to modify the binding thermodynamics of the immobilized cages within MOC@PILs. We infer that the alteration of the binding affinity for the immobilized cage results from the distinct microenvironments: The immobilized cage was surrounded by

cationic PIL chains and may become constrained compared to the free cage, while this effect is alleviated if the constraint around the cage loosens.

MS, Page 17:

Importantly, the swelling of MOC@PILs modified both the kinetics and thermodynamics of guest uptake, and higher degrees of swelling enabled the immobilized cages to bind guests similarly to the soluble cages.

Supplementary Information, Page S36:

To evaluate the influence of swelling of MOC@PILs on guest binding kinetics, we monitored the ratio of $[HG]/[H]_0$ as a function of time, where $[HG]$ and $[H]_0$ respectively represent the concentrations of the host-guest complex and the initial free host (*i.e.* total host), through ^1H NMR measurement. For the measurement of the binding kinetic curves of the soluble cage, a concentration of 1 mM cage in a 0.5 mL guest-saturated D_2O solution was used and the ^1H NMR spectra were measured after periods of time. For the measurement of the binding kinetic curves of the immobilized cage, the same amount of cage within MOC@PIL **6** as for the soluble cage was calculated and used. Similarly, MOC@PIL **6** was added into a 0.5 mL guest-saturated D_2O solution. After stirring for a period of time, the solid was separated through centrifugation and was washed with water three times to remove the surface-attached molecules of free guest. The release of cage species from MOC@PIL **6** into solution, including both guest-MOC and empty cage, was achieved by adding an excess of NaNO_3 and the corresponding ^1H NMR spectrum was recorded. Three guests, benzene, cyclohexane, and norbornane, were respectively investigated following the methods described above. Note that the binding of benzene with whichever the soluble or the immobilized cage was treated at rt, while the binding of the other two larger guests were conducted at 50°C to facilitate the binding equilibration.

Supplementary Figure 46. Binding kinetics of the soluble cage and swollen MOC@PIL **6** for benzene (a), cyclohexane (b), and norbornane (c). Binding kinetics of the swollen MOC@PIL **6** for benzene, cyclohexane, and norbornane (d).

We also monitored the concentration decrease of the free guests in the presence of swollen MOC@PILs to reveal the guest uptake kinetics. For benzene, cyclohexane, and norbornane, a concentration of 10 ppm in 20 mL water was prepared individually, and MOC@PIL **2**, **4**, or **6** was added into the aqueous solution for guest uptake (the molar ratio between immobilized cage and guest was 1.5 in each case). After stirring for periods of time, GC instrument equipped with a DB-WAX UI column was used to monitor the concentration of the guest in the solution. Note that the uptake of benzene with the immobilized cage was treated at rt, while the uptake of the other two larger guests were conducted at 50 °C to facilitate the binding equilibration.

Supplementary Figure 47. Removal efficiencies of MOC@PILs **2**, **4**, **6** for benzene (a), cyclohexane (b), and norbornane (c). Removal efficiencies of MOC@PIL **6** for benzene, cyclohexane, and norbornane (d).

8. The binding affinities in cage are ok (as they've been done before), but the apparent binding constants of cage-PIL are only estimates, as the authors state that not all the cage is released from the PIL. In addition, the concentration of free guests, many of which are volatile, could change upon centrifugation – this makes the accuracy of the numbers shown in S-Table 3 very weak. Certainly they should not be stated to 3 sig figs, and should only be described as estimates. The chart in the text (Fig 5) is probably ok, as all the affinities are relative and any errors should be constant. But I couldn't find out how the authors performed their error treatment, either. Overall, the S-Table affinities for cage:PIL should be far more tentative in their descriptions, and not promise accuracy that is impossible.

Reply: Although we have done our best to carefully perform the binding experiments and each experiment has been repeated at least once or twice, we acknowledge the unavoidable experimental error due to the complicated experimental procedures. We initially stated that “not all the cage is released from the PIL”, while we think this fact does not affect the accuracy of K_a in theory. As presented in Supplementary Section 5.3, based upon the following equation, the apparent K_a could be determined through the concentration of free guest [G] and the ratio of [HG]/[H]:

$$K_a = \frac{[HG]}{[H][G]}$$

The concentration of free guest [G] could be determined from integration of the NMR peaks relative to those of the *tert*-butanol internal standard directly, while the ratio of [HG]/[H] could be known by adding NO_3^- . Although the addition of NO_3^- was unable to exchange all the cage species on the polymer chains into the solution, the ratio between the released guest-MOC and the released empty cage in solution, *i.e.* [HG]/[H], should be the same as the initial ratio on the polymer chains. The apparent K_a determined in this way should be thus reliable in theory. However, as mentioned by the reviewer, the volatility of the guests as well as the complicated procedures for the measurement should affect the accuracy of the values. We have thus modified the numbers in these Supplementary Tables to make them more accurate and less ambiguous by reducing the significant figures. Moreover, the information on the repeat of experiments to calculate experimental error has been also added into the revised caption of Fig. 5.

Supplementary Information, Page S39:

Supplementary Table 3. Binding constants (K_a) of the soluble cage and apparent binding constants (apparent K_a) of the immobilized cage within MOC@PIL **6** for various guests in D_2O .

guest	K_a (M^{-1})	apparent K_a (M^{-1}) ^b
CHCl_3	2.43×10^3	2.7×10^3
benzene	2.89×10^3	2.7×10^3
1,4-cyclohexadiene	2.22×10^4	1.5×10^4
cyclohexene	1.43×10^4	6.9×10^3
cyclohexane	4.77×10^4	4.1×10^3
norbornadiene	1.25×10^4	3.4×10^3
norbornene	6.22×10^3	n.d. ^a
norbornane	2.29×10^3	3.8×10^2
7-oxabicycloheptane	2.71×10^2	3.4×10^2

^a The apparent binding constant could not be determined due to the difficulty in the assignment of peaks of the encapsulated guest on the ^1H NMR spectrum.

^b Due to the complicated procedures of the measurement, data are presented with only two significant figures to avoid inaccuracies. All errors are below 10%.

MS, Page 13:

Fig. 5 | Guest binding affinities of MOC@PILs. (a) Normalized K_a of the $\text{Fe}^{\text{II}}_4\text{L}_6$ cage and the apparent K_a of MOC@PIL **6** for a series of guests with differing sizes in water. (b) Apparent K_a of MOC@PILs **1-6** for benzene and norbornane in water. The dashed green and blue lines represent the binding constants of the soluble cage for benzene and norbornane, respectively. Error ranges were calculated from triplicate experiments.

9. The 2D NMRs are ok, but messy, and should be shown as contour plots, not bitmaps. The contours cannot be seen as shown.

Reply: Following the suggestion from the reviewer, we have revised all 2D NMR figures. All have now been re-plotted at higher resolution, with contours shown more clearly.

Supplementary Information, Page S17:

Supplementary Figure 23. ^1H - ^1H COSY spectrum (D_2O , 400 MHz, 298 K) of norbornadiene-MOC.

Supplementary Figure 24. ^1H - ^1H NOESY spectrum (D_2O , 400 MHz, 298 K) of norbornadiene@MOC. The NOE peaks between the encapsulated guest protons and occupied cage have been highlighted.

Supplementary Figure 26. ^1H - ^1H COSY spectrum (D_2O , 400 MHz, 298 K) of norbornene-MOC.

Supplementary Figure 27. ^1H - ^1H NOESY spectrum (D_2O , 400 MHz, 298 K) of norbornene@MOC. The NOE peaks between the encapsulated guest protons and occupied cage have been highlighted.

Supplementary Figure 29. ^1H - ^1H COSY spectrum (D_2O , 400 MHz, 298 K) of norbornane-CMO.

Supplementary Figure 30. ^1H - ^1H NOESY spectrum (D_2O , 400 MHz, 298 K) of norbornane@MOC. The NOE peaks between the encapsulated guest protons and occupied cage have been highlighted.

Supplementary Figure 32. ^1H - ^1H COSY (D_2O , 400 MHz, 298 K) spectrum of 7-oxabicycloheptane-MOC.

Supplementary Figure 33. ^1H - ^1H NOESY (D_2O , 400 MHz, 298 K) spectrum of 7-oxabicycloheptane-CMOc. The NOE peaks between the encapsulated guest protons and occupied cage have been highlighted.

We thank again to Reviewer 1 for these very helpful suggestions and we have done our best to address them. We believe the quality of the manuscript has been greatly improved, and if possible, we are very happy to hear more comments or suggestions from the reviewer.

Referee: 2

In this article, the authors report the preparation and application of a hybrid material constructed from a poly(ionic liquid) (PIL) and a known negatively charged metal-organic cage (MOC). Different loadings of MOC to PIL were explored, and the materials and swelling were characterized using a number of techniques. The applications of the hybrid material in micropollutant removal from water were explored; additionally, the authors have reported novel host-guest chemistry for this MOC and used this information to extend the scope of the micropollutant removal studies.

I think this is a very clever approach to the construction of complex materials based on simple underlying principles. For me it is a clear step forward from existing work and is thus novel enough for publication in *Nat. Comm.* However, certain areas of discussion in this require further consideration before publication is recommended.

Reply: We appreciate the positive comments from the reviewer as well as the very helpful suggestions in the report. We have carefully addressed all the issues raised by the reviewer, as detailed below, and are very looking forward to hearing the opinion from the reviewer again.

1. The discussion in the introduction focusses on the applications of MOCs in the solution and solid-state, but some attention should be given to existing examples that bridge this gap to provide fuller context/precedent for this work. The incorporation of MOCs into polymer hydrogels is certainly relevant here, and although several references have been included to highly relevant manuscripts in this area, I think the introduction must explicitly describe this precedent. I think that the incorporation of MOCs into porous liquids is also very relevant. A recent manuscript describing permanently porous ionic liquid gels based on metal-organic polyhedra (DOI: 10.1021/jacs.3c03778) should be acknowledged.

Reply: We agree with the reviewer that the incorporation of MOCs into gels is highly relevant to the work, and this part of knowledge should be documented in the Introduction. Following the suggestion from the reviewer, we have added discussion on MOC-integrated gels, including the current advances and common challenges in this area, into the revised Introduction of the MS. Relevant references, including the recent paper describing permanently porous ionic liquid gels, have been also cited and highlighted in the revised MS.

MS, Page 3:

The unique positioning of gel materials at the boundary between liquids and solids offers attractive features when combining with MOCs.⁴⁸ Particular attention has been devoted to the coupling of polymers with MOCs to prepare MOC-branched (star) or crosslinked networks.^{49,50} The MOC-integrated gels have shown tailored mechanical properties, self-healing ability, switchable network topology, and even permanent porosity.⁵¹⁻⁵⁴ It is envisioned that the confined solvent within the gel could provide ideal microenvironments for the coordination cages to dynamically move and flex, enabling these cages to bind guests efficiently. Similarly to hard solids, capsule-containing soft solids can also be recycled in a heterogeneous fashion. However, to our knowledge, the host-guest chemistry of these integrated MOCs is rarely explored.⁵¹ Moreover, the strategy to prepare MOC-integrated gels is currently limited to the use of MOCs as junctions.

MS, Page 21:

48. Jahović, I., Zou, Y.-Q., Adorinni, S., Nitschke, J.R. & Marchesan, S. Cages meet gels: Smart materials with dual porosity. *Matter* 4, 2123-2140 (2021).

49. Hosono, N. & Kitagawa, S. Modular design of porous soft materials via self-organization of metal-organic cages. *Acc. Chem. Res.* **51**, 2437-2446 (2018).
50. Liu, J., *et al.* Self-healing and shape memory hypercrosslinked metal-organic polyhedra polymers via coordination post-assembly. *Angew. Chem. Int. Ed.* **61**, e202212253 (2022).
51. Gu, Y., *et al.* Photoswitching topology in polymer networks with metal-organic cages as crosslinks. *Nature* **560**, 65-69 (2018).
52. Zhukhovitskiy, A.V., *et al.* Highly branched and loop-rich gels via formation of metal-organic cages linked by polymers. *Nat. Chem.* **8**, 33-41 (2016).
53. Wang, Z., *et al.* Pore-networked gels: Permanently porous ionic liquid gels with linked metal-organic polyhedra networks. *J. Am. Chem. Soc.* **145**, 14456-14465 (2023).
54. Wang, Y., *et al.* Block co-polyMOCs by stepwise self-assembly. *J. Am. Chem. Soc.* **138**, 10708-10715 (2016).

2. The authors state they used ICP-AES to determine the amount of the MOC immobilized in the PIL after digestion with nitric acid. I believe that this assay can detect the amount of Fe immobilized in each sample, not the amount of in-tact MOC (which is a difficult question to address). This quantification does not account for the possibility that some of the cage within the PIL could have decomposed before the digestion step.

The next experiment, in which the cage is released from the PIL after ion exchange and analysed by ¹H NMR goes some way to addressing this – however, it doesn't seem to be quantitative and would not be able to detect any non-coordinated Fe present. We still do not know what the loading of in-tact cage in each sample is. Later experiments (FTIR, ¹³C MAS NMR and TGA) show increased loadings in the different samples but cannot give an absolute loading.

Reply: We appreciate the very good comment from the reviewer, which is also a difficult but important issue to address. Regarding the hypothetical presence of non-coordinated Fe^{II} in the MOC@PILs samples, it would be unfavorable for cationic Fe^{II} to be integrated into a cationic PIL whereas the Fe^{II}₄L₆ cage is anionic and interacts favorably with the cationic PIL. The Fe^{II} content detected by ICP-AES thus most likely originates from the cage, or at worst, from Fe-coordinated complexes. After in-depth consideration, we have designed and conducted two sets of new experiments to test the stability of the MOC. These include the stability of the MOC when in the presence of a large excess of PIL monomers and the state of the solution cage after immobilized by PIL-NO₃⁻. As detailed below, the results of the experiments demonstrate the high stability of the cage - no decomposition species have been observed in either case. This guarantees that the amount of the detected Fe by ICP-AES exclusively comes from the intact immobilized MOCs. These experiments as well as the corresponding discussion have been added into the revised MS and the revised Supplementary Information.

MS, Page 6:

Through simple stirring of the synthesized PIL with the Fe^{II}₄L₆ cage in water, ion exchange took place, resulting in obvious color change of the material from colorlessness to purple (Fig. 2a). The precipitate was obtained by centrifugation and was washed with pure water repeatedly until no color resulting from the free Fe^{II}₄L₆ cage was observed in the supernatant. The addition of a large amount of acetone enabled the cage-gel composite to agglomerate. The amount of the Fe^{II}₄L₆ cage in the composite could be quantified by ICP-AES for the measurement of Fe after digesting the sample with nitric acid. We infer the amount of the detected Fe to be exclusively from the intact immobilized MOCs due to the high stability of the Fe^{II}₄L₆ cage based on the following two sets of experiments: (i) ¹H

NMR experiments indicated the integrity of the $\text{Fe}^{\text{II}}_4\text{L}_6$ cage in D_2O without any decomposition in the presence of a large excess of PIL monomers (12 equiv.) (Supplementary Fig. 6); (ii) After saturating PIL-NO_3^- with an excess of cage for immobilization, only pure cage could be observed in the solution with no trace of decomposition species, such as any subcomponents (Supplementary Fig. 7). Samples (including MOC@PILs **1-6**) with varying loadings of the $\text{Fe}^{\text{II}}_4\text{L}_6$ cage from 0.11 to 0.74 g/g were thus prepared (see Supplementary Section 2.3).

Supplementary Information, Page S7:

Prior to the synthesis of MOC@PILs , we conducted experiments to confirm the integrity and stability of the $\text{Fe}^{\text{II}}_4\text{L}_6$ cage in the presence of PIL-NO_3^- . After mixing the $\text{Fe}^{\text{II}}_4\text{L}_6$ cage with a large excess of PIL monomers in D_2O (12 equiv. $[\text{VEIM}]\text{NO}_3$ relative to the cage; the ratio between the two is higher than that of the integrated MOC and the monomer within MOC@PIL **1**), the ^1H NMR spectrum of the sample were measured (Supplementary Figure 6). Compared with the ^1H NMR spectrum of the $\text{Fe}^{\text{II}}_4\text{L}_6$ cage, no noticeable variation was observed when in the presence of 12 equiv. $[\text{VEIM}]\text{NO}_3$.

Supplementary Figure 6. ^1H NMR (D_2O , 400 MHz, 298 K) spectra of the $\text{Fe}^{\text{II}}_4\text{L}_6$ cage, the $\text{Fe}^{\text{II}}_4\text{L}_6$ cage in the presence of 12 equiv. $[\text{VEIM}]\text{NO}_3$, and $[\text{VEIM}]\text{NO}_3$.

Supplementary Information, Page S8:

The supernatants were monitored using ^1H NMR spectroscopy during the synthesis of MOC@PILs . As nondeuterated water was used during the synthesis, a capillary filled with D_2O was inserted into the NMR tube containing the supernatant sample for field frequency locking (Supplementary Figure 7).

Supplementary Figure 7. ^1H NMR (400 MHz, 298 K) spectra of the supernatants obtained from the complete swelling of 0.5 g PIL-NO_3^- in 200 mL water (a), thorough mixing of 0.5 g PIL-NO_3^- and 1.0 g cage in 200 mL water (b), thorough mixing of 0.5 g PIL-NO_3^- and 2.0 g cage in 200 mL water (c), and the spectrum of the $\text{Fe}^{\text{II}}_4\text{L}_6$ cage (d). A capillary filled with D_2O was inserted into the NMR tube containing the supernatant sample for field frequency locking.

3. In the section on micropollutant removal – experiments to monitor the removal of CH_2Cl_2 , CHCl_3 , benzene, 1,4-cyclohexadiene, cyclohexane, norbornadiene, norbornene and norbornane from D_2O were conducted by NMR and the authors state the removal efficiency was “close to 100%”. I would be cautious of using NMR experiments to validate such claims given the low sensitivity compared to other techniques such as GC (which is used later). What’s the minimum concentration of these micropollutants that can actually be detected using such experiments? The number of scans performed should be reported and the authors should acknowledge the limitations of NMR in their discussion.

Additionally, in the SI, the authors state “The concentration of the micropollutant after adsorption was analyzed by ^1H NMR spectroscopy, and tert-butanol or ethylene glycol was used as the internal standard”. However, no concentrations calculated in this way or the exact removal efficiencies are actually reported – so we cannot determine how “close to 100%” the efficiencies really are.

Reply: We agree with the reviewer that we should be cautious when we made a conclusion of “close to 100%” of removal efficiency if NMR technique was used. In order to know the sensitivity of our NMR equipment and accurately calculate the removal efficiency, benzene samples at various concentrations (22.9, 10, 1, 0.1, and 0.05 mM) in D_2O were measured by the same piece of NMR equipment with the same number of scans (64 scans) for each measurement. Results showed that the signal of 0.05 mM benzene, equivalent to “3 mM C-H”, in D_2O was still visible on the ^1H NMR spectrum. Based upon the probed sensitivity, the removal rates of the organic pollutants with no more proton signal appeared after adsorption were recalculated and tabulated in the revised Supplementary Table 5. For the removal of 1,4-dioxane and 7-oxabicycloheptane, we could still use the internal standard to calculate the removal rates due to the presence of obvious proton signals after adsorption. We have added these results into the revised MS and the revised Supplementary Information. Moreover, the number of scans in each ^1H NMR experiment have been also indicated in the revised captions of the relevant figures.

After thorough mixing and equilibration, the swollen (pollutant@MOC)@PIL **6** was separated from the purified water *via* centrifugation. ^1H NMR spectra of the purified D_2O indicated that pollutants CH_2Cl_2 , CHCl_3 , benzene, and 1,4-cyclohexadiene were removed efficiently ($\geq 94\%$) after 2 h at room temperature (rt) (Supplementary Figs. 48-51). Besides, almost complete removal of cyclohexene, cyclohexane, norbornadiene, norbornene, and norbornane from water were also exhibited after 6 h at 50°C (Supplementary Figs. 53-57).

Supplementary Information, Page S46:

For the removal of CH_2Cl_2 , CHCl_3 , benzene, 1,4-cyclohexadiene, cyclohexene, cyclohexane, norbornadiene, norbornene, and norbornane from water, almost no proton signal of guests could be observed after adsorption by MOC@PIL **6**. In order to accurately calculate the removal efficiency, the sensitivity of the ^1H NMR technique should be known. Benzene samples at various concentrations (22.9, 10, 1, 0.1, and 0.05 mM) in D_2O were thus measured by the same piece of NMR equipment with the same number of scans (64 scans) at 298 K for each sample. As shown in Supplementary Figure 59, the signal of 0.05 mM benzene, equivalent to “3 mM C-H”, in D_2O was still visible on the ^1H NMR spectrum. Based upon the probed sensitivity, the removal rates of these organic pollutants were calculated and tabulated in Supplementary Table 5. It should be noted that as no signal of guests could be observed after adsorption by MOC@PIL **6** on the ^1H NMR spectra, the pollutant removal data were underestimated and only the lower limits are provided.

For the removal of 1,4-dioxane and 7-oxabicycloheptane from water, removal rates were calculated directly based upon the relative intensities of the signals of the guests and the *tert*-butanol internal standard.

Supplementary Figure 59. ^1H NMR (D_2O , 400 MHz, 298 K, number of scans = 64) spectra of benzene at various concentrations (22.9 mM, 10 mM, 1 mM, 0.1 mM and 0.05 mM) in D_2O .

Supplementary Table 5. Removal rates (%) of the organic pollutants from D₂O by adding MOC@PIL 6.

guest	removal rate (%)
CH ₂ Cl ₂	≥ 97
CHCl ₃	≥ 94
benzene	≥ 99
1,4-cyclohexadiene	≥ 98
1,4-dioxane	25
cyclohexene	≥ 98
cyclohexane	≥ 99
norbornadiene	≥ 98
norbornene	≥ 98
norbornane	≥ 99
7-oxabicycloheptane	45

Supplementary Information, Page S40:

Supplementary Figure 48. ¹H NMR (D₂O, 400 MHz, 298 K, number of scans = 64) spectra of the polluted water before and after adsorption of CH₂Cl₂. The initial concentration of CH₂Cl₂ was 5 mM.

Supplementary Figure 49. ¹H NMR (D₂O, 400 MHz, 298 K, number of scans = 64) spectra of the polluted water before and after adsorption of CHCl₃. The initial concentration of CHCl₃ was 5 mM.

Supplementary Figure 50. ¹H NMR (D₂O, 400 MHz, 298 K, number of scans = 64) spectra of the polluted water before and after adsorption of benzene. The initial concentration of benzene was 5 mM.

Supplementary Figure 51. ¹H NMR (D₂O, 400 MHz, 298 K, number of scans = 64) spectra of the polluted water before and after adsorption of 1,4-cyclohexadiene. The initial concentration of 1,4-cyclohexadiene was 5 mM.

Supplementary Figure 52. ¹H NMR (D₂O, 400 MHz, 298 K, number of scans = 64) spectra of the polluted water before and after adsorption of 1,4-dioxane. The initial concentration of 1,4-dioxane was 5 mM. The concentration of 1,4-dioxane was reduced to 3.75 mM after adsorption with MOC@PIL 6.

Supplementary Figure 53. ¹H NMR (D₂O, 400 MHz, 298 K, number of scans = 64) spectra of the polluted water before and after adsorption of cyclohexene. The initial concentration of cyclohexene was 5 mM.

Supplementary Figure 54. ¹H NMR (D₂O, 400 MHz, 298 K, number of scans = 64) spectra of the polluted water before and after adsorption of cyclohexane. The initial concentration of cyclohexane was 5 mM.

Supplementary Figure 55. ¹H NMR (D₂O, 400 MHz, 298 K, number of scans = 64) spectra of the polluted water before and after adsorption of norbornadiene. The initial concentration of norbornadiene was 5 mM.

Supplementary Figure 56. ¹H NMR (D₂O, 400 MHz, 298 K, number of scans = 64) spectra of the polluted water before and after adsorption of norbornene. The initial concentration of norbornene was 1.34 mM.

Supplementary Figure 57. ^1H NMR (D_2O , 400 MHz, 298 K, number of scans = 64) spectra of the polluted water before and after adsorption of norbornane. The initial concentration of norbornane was 1.46 mM.

Supplementary Figure 58. ^1H NMR (D_2O , 400 MHz, 298 K, number of scans = 64) spectra of the polluted water before and after adsorption of 7-oxabicycloheptane. The initial concentration of 7-oxabicycloheptane was 5 mM. The concentration of 7-oxabicycloheptane was reduced to 3.33 mM after adsorption with MOC@PIL 6.

Referee: 3

This article describes the fabrication of new composite materials based on water-swallowable poly(ionic liquid)s (PILs) and coordination cages for guest capture and separation. The concept is very interesting to use the swelling PILs to provide a solution-like microenvironment for the cages, which allows the cages to be dynamic and bind guests, followed by the deswelling process for recycling. In reading the abstract and introduction of the manuscript, I had high expectations for the work, but it turned out to be overselling and lacked mechanistic studies. Therefore, I do not recommend this current version be accepted in *Nature Communications*. The authors should address the following severe scientific concerns.

Reply: We appreciate both the positive and critical comments from the reviewer, as well as the very good suggestions in the report. The reviewer has raised up many helpful suggestions, especially in the aspects of gel characterization and mechanistic studies. To address these concerns, we have supplied several experiments and thus particularly added a new section, “**Swellability, mechanical properties, and morphology of MOC@PILs**”, into the revised MS. A detailed list of corrections is presented below. We are also very happy to further improve the manuscript if we can hear more suggestions from the reviewer.

1. The authors attributed the change of cage@PILs swelling capacity to be the outcome of (a) cage hydrophobicity and (b) the limited motion of PIL chains due to their electrostatic interaction with cages. As far as I understand, the cages with four negative charges can serve as crosslinkers to more densely link the PIL chains to prevent their swelling in water. However, the authors did not demonstrate any experiments to support this hypothesis. For instance, the authors could have demonstrated the mesoscale characterization of composites after freeze-drying. However, the only cage@PIL 4 was characterized by SEM. The authors should check all the mesoscale structures with different cage loading and discuss how the resulting morphology, size, and width would be affected by the loading amount. The authors should show evidence to support their hypothesis.

Reply: We appreciate the great suggestion from the reviewer and fully agree with the reviewer that cages with four negative charges can serve as crosslinkers to more densely link the PIL chains to prevent their swelling in water. This explanation has been added into the revised MS along with the addition of experimental evidences. Following the suggestion from the reviewer, apart from MOC@PIL 4, we have also conducted measurements of Cryo-SEM of swollen MOC@PILs 2 and 6 after freeze-drying for comparison. Results demonstrated a significant effect of cage loading on the morphology and pore size of swollen MOC@PILs. The figures and the corresponding discussion have been added into the revised MS.

MS, Page 7:

We infer this phenomenon to have two main causes: 1) the anionic cages are more hydrophobic than nitrate anions, which reduces the affinity of the composite with water; 2) each cage with four negative charges requires to be surrounded by four imidazolium cations

for charge balance, serving as a crosslinker to more densely link the cationic PIL chains to prevent their swelling in water.

MS, Page 8:

Fig. 3 | Swelling capacity, mechanical properties, and morphology of swollen MOC@PILs. (a) Swelling capacity and storage moduli G' of MOC@PILs **1-6** and the parent PIL-NO₃⁻. (b) Storage and loss moduli, G' and G'' , of the swollen PIL-NO₃⁻ and MOC@PILs **2, 4**, and **6** as a function of angular frequency ω . (c) Photographs of PIL-NO₃⁻ and MOC@PILs **1-6** before and after swelling. (d-g) Cryo-SEM images of swollen PIL-NO₃⁻ and MOC@PILs **2, 4** and **6** after freeze-drying showing the microstructures.

MS, Page 9:

The Cryo-SEM images of swollen MOC@PILs after freeze-drying revealed interconnected porous network of the gels (Figs. 3d-g), in contrast to the initial agglomerate state prior to swelling (Supplementary Fig. 18). Akin to the microstructure of the parent PIL-NO₃⁻, expanded pores are present in a regular layout within the swollen MOC@PILs, while the diameters of which continuously decreased from **2** (30–50 μm) to **4** (10–30 μm) and **6** (3–6 μm). This result was consistent with the observation of macroscopic volume contraction from **2** to **6** (Fig. 3c), and demonstrated the significant effect of cage loading on the morphology and pore size of swollen MOC@PILs. The presence of the large honeycomb-like pores, even for the swollen MOC@PIL **6** having the highest cage loading, is beneficial for mass transfer and cage accessibility, ensuring sufficient contact of the anionic cages with substrates for host-guest interactions.

2. Related to the above discussion, the authors should also carry out the macroscopic characterization such as gel rheology (viscoelastic property). The quantitative analysis of mechanical properties can be correlated to the interaction between PIL and the cages. The authors only mention: “The maximum cage loading led to a swelling ratio of 5 for cage@PIL **6** (0.74 g/g), the texture of which was relatively stiff.” The authors should quantitatively characterize this stiffness by a conventional rheological method.

Reply: We agree with the reviewer that quantitative analysis of the stiffness of the swollen MOC@PILs gels are very important. Following the suggestion from the reviewer, rheology studies of the gels have been conducted. Results showed that the gradual increase of the cage loading (> 0.32 g/g) within MOC@PILs led to the concomitant increase of the storage moduli, suggesting the increased stiffness. The detailed results, figures, and the corresponding discussion have been added into the revised MS and the revised Supplementary Information.

MS, Page 9:

The decrease of swelling capacity and enhancement of gel stiffness with increasing MOC loadings were confirmed through rheological studies. Frequency sweep tests at 0.1% shear strain showed the storage moduli (G') dominated over the loss moduli (G'') for all the swollen gels within the tested range, indicating strong elastic response from all the samples (Fig. 3b and Supplementary Fig. 16). The G' value of hydrogel **1** was 103 Pa at an oscillatory frequency of 10.05 rad s^{-1} , close to the value of the parent PIL- NO_3^- ($G' = 89$ Pa) (Fig. 3a). In contrast, the gradual increase of the cage loading (> 0.32 g/g) within MOC@PILs witnessed concomitant increase of G' , suggesting the increased stiffness. A G' of 1.01×10^6 Pa was obtained for MOC@PIL **6** with a maximum cage loading of 0.74 g/g, corresponding to a 10^4 -fold increase with respect to the parent PIL- NO_3^- . These rheology results agree with the tests of swelling capacity discussed above, revealing the role of crosslinker played by the immobilized cages.⁵⁴ Moreover, time-dependent oscillatory tests demonstrated high stability of all these swollen gels (Supplementary Fig. 17).

MS, Page 8:

Fig. 3 | Swelling capacity, mechanical properties, and morphology of swollen MOC@PILs. (a) Swelling capacity and storage moduli G' of MOC@PILs **1-6** and the parent PIL- NO_3^- . (b) Storage and loss moduli, G' and G'' , of the swollen PIL- NO_3^- and MOC@PILs **2, 4, and 6** as a function of angular frequency ω . (c) Photographs of PIL- NO_3^- and MOC@PILs **1-6** before and after swelling. (d-g) Cryo-SEM images of swollen PIL- NO_3^- and MOC@PILs **2, 4 and 6** after freeze-drying showing the microstructures. (please see the revised figure for Q1)

MS, Page 17:

We have proposed for the first time the potential of using swelling of polymers to control the flexibility, dynamicity, and even guest binding mechanisms of coordination cages. This concept was demonstrated by integration of anionic cages into swellable cationic PILs through ion exchange, resulting in a series of MOC@PILs having differing cage loadings. The amount of the immobilized cage within MOC@PILs were found to control the swelling ability and mechanical properties of MOC@PILs. In comparison to the conventional modulation of MOC junctions to tune the mechanical properties of MOCs-branched or crosslinked gels,^{48,49} the ion exchange strategy developed here was simpler and more straightforward.

MS, Page 21:

48. Jahović, I., Zou, Y.-Q., Adorinni, S., Nitschke, J.R. & Marchesan, S. Cages meet gels: Smart materials with dual porosity. *Matter* **4**, 2123-2140 (2021).

49. Hosono, N. & Kitagawa, S. Modular design of porous soft materials via self-organization of metal-organic cages. *Acc. Chem. Res.* **51**, 2437-2446 (2018).

54. Wang, Y., *et al.* Block co-polyMOCs by stepwise self-assembly. *J. Am. Chem. Soc.* **138**, 10708-10715 (2016).

Supplementary Information, Page S13:

Supplementary Figure 16. Frequency sweeps in oscillatory rheology of the swollen PIL-NO_3^- and MOC@PILs 1-6 ranging from 0.1 to 100 rad s^{-1} at a 1.0% strain amplitude.

Supplementary Figure 17. Time-dependent oscillatory curves of the swollen PIL-NO_3^- and MOC@PILs 1-6 at a 0.1% strain amplitude.

- The authors claimed on page 12 as follows: “These results demonstrate the important effect of swelling on guest binding: The higher degree of swelling allows immobilized cage to be more flexible and dynamic to present binding performances similar to the level under solution-state conditions even for larger guests”. I still do not understand this statement. The authors should separately consider the macroscopic swelling behavior from the microscopic crosslinking between PILs and cages. The former should influence the water uptake within the gels, which means the hydration of the cages and concentration of the cages. The latter is truly the electrostatic interaction between PILs and the cages. The authors should consider how the swelling process would influence the interaction between PILs and cages. Is it possible to determine this interaction by carefully analyzing any NMR technique?

Reply: As suggested by the reviewer, we should separately consider the macroscopic swelling behavior from the microscopic crosslinking between PILs and cages, and it is very important to know the electrostatic interactions between PILs and cages. As answered for Q1 and Q2, we have supplied experiments of SEM and rheology studies to investigate the cage effect on macroscopic swelling behavior. Following the suggestion from the reviewer, we have also conducted ^1H NMR experiments to demonstrate the existence of electrostatic interactions between PILs and cages, as detailed below.

Results showed that for the ^1H NMR spectra of swollen MOC@PILs **1-6**, the increase of the cage loading from gels **1** to **6** afforded gradual upfield shifts of proton signals of the polymer chains, consistent with the expected interactions. Moreover, unnoticeable changes between ^1H NMR spectra of **1** at different swelling degrees were observed, indicating the swelling degree was unable to perturb the strength of the interactions. The detailed results, figures, and the corresponding discussion have been added into the revised MS and the revised Supplementary Information.

MS, Page 8:

To demonstrate the electrostatic interactions between the immobilized anionic cages and the cationic PIL chains, ^1H NMR spectra for the fully swollen gels of PIL- NO_3^- and MOC@PILs **1-6** were recorded. Results showed that with the increase of the cage loading from gels **1** to **6**, gradual upfield shifts of proton signals of the imidazolium and ethyl groups were observed, consistent with the expected interactions (Supplementary Fig. 14). We also measured the ^1H NMR spectra of MOC@PIL **1** at different swelling degrees (50, 100, 200, 317-fold) by adding differing amounts of water. Unnoticeable changes between spectra of **1** were observed, indicating the swelling degree was unable to perturb the strength of the interactions between the immobilized MOCs and the PIL chains (Supplementary Fig. 15).

Supplementary Information, Page S12:

^1H NMR spectra of the fully swollen MOC@PILs **1-6** and the parent PIL- NO_3^- were measured to investigate the electrostatic interactions between the immobilized cages and the PIL chains of MOC@PILs (Supplementary Figure 14). Moreover, for the same purpose, ^1H NMR spectra of MOC@PIL **1** at different swelling degrees (50, 100, 200, 317-fold) by adding differing amounts of water were also measured (Supplementary Figure 15).

Supplementary Figure 14. ^1H NMR (D_2O , 400 MHz, 298 K) spectra of monomer [VEIM] NO_3 , the fully swollen gels of PIL- NO_3^- and MOC@PILs **1-6**, and the Fe^{14}L_6 cage.

Supplementary Figure 15. ^1H NMR (D_2O , 400 MHz, 298 K) spectra of MOC@PIL **1** at different swelling degrees (50, 100, 200, 317-fold). The peak of acetone originally left within the NMR tubes for cleaning is labelled.

4. Related to the above, the strange behavior was observed in Fig. 3a. This is not a linear correlation. But there is a threshold at the lower concentration domain of the cage loading. What about the concentration lower than 0.32 g/g for cage@PIL **1**? It seems a certain concentration is essential for the crosslinking of PILs by the cage molecules to influence the swelling property.

Reply: Following the suggestion from the reviewer, we have also prepared two additional samples of cage@PILs with cage loadings lower than 0.32 g/g (0.11 and 0.24 g/g). Swelling experiments suggested the swelling capacities of 327 and 322 for the two samples, which are very close to the values of the parent PIL- NO_3^- ($Q = 350$) and MOC@PIL **1** ($Q = 317$, cage loading 0.32 g/g). As predicted by the reviewer, these results demonstrated that a certain concentration of the immobilized cage is essential for the crosslinking of PILs to influence the swelling property. These results and the corresponding discussion have been added into the revised MS.

MS, Page 7:

Agglomerate samples of MOC@PILs could rapidly swell in water and reached equilibrium within 20 min (Supplementary Fig. 13), resulting in a series of purple hydrogels. Importantly, the swelling capacity of MOC@PILs could be adjusted by altering the loadings of the immobilized cages. When the cage loading was lower than 0.32 g/g, the swelling capacity of MOC@PILs in water was not significantly altered. For instance, the swelling capacity of MOC@PIL **1** ($Q = 317$, cage loading, 0.32 g/g) was very close to that of the parent PIL- NO_3^- ($Q = 350$). Interestingly, upon reaching this threshold of cage loading (0.32 g/g), the values of Q were observed to decrease almost linearly from MOC@PIL **1** to **6** (Figs. 3a and 3b). A maximum cage loading of 0.74 g/g for MOC@PIL **6** led to a swelling capacity of only 5.

MS, Page 8:

Fig. 3 | Swelling capacity, mechanical properties, and morphology of swollen MOC@PILs. (a) Swelling capacity and storage moduli G' of MOC@PILs **1-6** and the parent PIL- NO_3^- . (please see the revised Fig. 3a for Q1)

Supplementary Information, Page S8:

The Fe contents of MOC@PILs were measured by ICP-AES after digesting the samples of MOC@PILs (40 mg) with concentrated nitric acid (2 mL) at 120 °C for 12 h. The amounts of the anionic cage within the eight samples of MOC@PILs were calculated to be 0.11, 0.24, 0.32 (MOC@PIL 1), 0.42 (MOC@PIL 2), 0.51 (MOC@PIL 3), 0.62 (MOC@PIL 4), 0.65 (MOC@PIL 5), and 0.74 (MOC@PIL 6) g/g, respectively, based on the amounts of Fe measured in each sample.

Supplementary Information, Page S11:

The swelling capacity were calculated to be 327, 322, 317 (MOC@PIL 1), 219 (MOC@PIL 2), 157 (MOC@PIL 3), 101 (MOC@PIL 4), 78 (MOC@PIL 5), and 5 (MOC@PIL 6), respectively.

5. The authors did not describe why cage@PIL 6 was chosen as the representative adsorbent for the micropollutant removal and purification. This sample possesses the lowest swelling capacity and can be easily recycled from the solution without the deswelling process. As far as I understand, the addition of acetone here is to help the release of the encapsulated organic molecules rather than the deswelling. To study the effect of sample swelling behavior, the authors should compare the pollutant extraction performance among cage@PILs with different cage loading amounts.

Reply: Following the suggestion from the reviewer, we have added the reasons why we chose MOC@PIL 6 as the representative adsorbent in the revised MS. MOC@PIL 6 was chosen as the adsorbent due to the high cage loading and appropriate stiffness after swelling, which could be easily recycled after adsorption. The low swellability of 6 ($Q = 5$) also adsorbed only a small amount of water for in-situ swelling, allowing a large proportion of water sample to be left. As suggested by the reviewer, we have also studied the effect of swelling of samples with different cage loadings on the adsorption kinetics and adsorption performance, results showed that all three samples of MOC@PILs 2, 4, 6 could achieve the complete removal of benzene, cyclohexane, and norbornane within a reasonable period of time. This new result, which has been added into the revised MS, constituted another important reason of using 6 as the represented adsorbent.

MS, Page 14:

Pollutant treatment is critical in modern society and adsorption is a leading technology for removing pollutants from water.^{62,63} The above results suggested the great potential of using swollen MOC@PILs as efficient and regenerable adsorbents for water purification. In this context, we chose MOC@PIL 6 as the adsorbent due to the high cage loading and appropriate stiffness after swelling, which could be easily recycled after adsorption through centrifugation or filtration. The low swellability of 6 ($Q = 5$) also consumed only a tiny amount of water for in-situ swelling, allowing a large proportion of water sample to be left. Moreover, the above adsorption kinetic experiments of 6 (Supplementary Fig 47) also indicated the capability of complete removal of benzene, cyclohexane, and norbornane within a reasonable period of time.

MS, Page 12:

Moreover, we also monitored the concentration decrease of the free guests (initially fixed at 10 ppm) when in the presence of swollen MOC@PILs (2, 4, 6; 1.5 equiv. of the immobilized cage relative to the guest). MOC@PILs with higher swellability were found to generally present faster guest removal (Supplementary Fig 47). These results reveal the effect of swelling on guest uptake kinetics: The higher degree of swelling allows immobilized cage to be more flexible and dynamic to present faster binding kinetics, while small guests, such as benzene, that are relatively easy to enter the cavity, could be bound rapidly even using MOC@PIL 6 that has the lowest swellability.

Supplementary Information, Page S37:

We also monitored the concentration decrease of the free guests in the presence of swollen MOC@PILs to reveal the guest uptake kinetics. For benzene, cyclohexane, and norbornane, a concentration of 10 ppm in 20 mL water was prepared individually, and MOC@PIL 2, 4, or 6 was added into the aqueous solution for guest uptake (the molar ratio between immobilized cage and guest was 1.5 in each case). After stirring for periods of time, GC instrument equipped with a DB-WAX UI column was used to monitor the concentration of the guest in the solution. Note that the uptake of benzene with the immobilized cage was treated at rt, while the uptake of the other two larger guests were conducted at 50 °C to facilitate the binding equilibration.

Supplementary Figure 47. Removal efficiencies of MOC@PILs 2, 4, 6 for benzene (a), cyclohexane (b), and norbornane (c). Removal efficiencies of MOC@PIL 6 for benzene, cyclohexane, and norbornane (d).

6. Regarding the extraction experiment of micropollutant, the sample of cage@PIL 6 showed complete removal of quite different types of molecules. Due to its low swelling capacity, the extraction of bigger molecules was shown to take longer time and require higher temperatures. The authors should show the kinetics adsorption data as a function of time and compare it with other cage@PILs to figure out the role of the swelling process of composites.

Reply: Following the suggestion from the reviewer, we have conducted kinetic adsorption experiments of using MOC@PILs having different swelling capacities for three guests, benzene, cyclohexane, and norbornane. As answered in Q5, MOC@PILs with higher swellability were found to generally present faster guest removal (Supplementary Fig 47), consistent with our expectation that the higher degree of swelling allows immobilized cage to be more flexible and dynamic to present faster binding kinetics. Nevertheless, all three samples of MOC@PILs 2, 4, 6 could achieve the

complete removal of benzene, cyclohexane, and norbornane within a reasonable period of time. (For details, please see the answer in Q5)

7. More importantly, the authors demonstrated the “micropollutant” removal experiments with very high concentrations of pollutants, which is not realistic for micropollutants. Indeed, the concentration was not shown in the main text and I needed to check the experimental sections in the supporting information. According to the SI, the concentration of micropollutants is fixed at 5 mM, which might be in the range of a few hundred ppm. These values are at least 5 orders of magnitude higher than the practical situation because a typical concentration range of organic micropollutants is between a few ppb and tens of ppt. The authors should not call this experiment “micropollutant” removal.

Reply: Many thanks to the reviewer for pointing this out and we apologize for our initial mistake. Following the suggestion from the reviewer, we have changed all the words of “micropollutant” to “pollutant” to be more accurate.

MS, Page 14:

Pollutant removal and chemical purification.

As shown in Fig. 6, agglomerate MOC@PIL 6 swelled in polluted water containing pollutants (the molar ratio between immobilized cage and pollutant was 1.2 or higher), and guest binding occurred. After thorough mixing and equilibration, the swollen (pollutant@MOC)@PIL 6 was separated from the purified water *via* centrifugation. ¹H NMR spectra of the purified D₂O indicated that pollutants CH₂Cl₂, CHCl₃, benzene, and 1,4-cyclohexadiene were removed efficiently (≥ 94%) after 2 h at room temperature (rt) (Supplementary Figs. 48-51).

1,4-Dioxane and 7-oxabicycloheptane, the two polar organic pollutants, could be also captured, although their efficiencies were lower than others (25% and 45% for 1,4-dioxane and 7-oxabicycloheptane, respectively) (Supplementary Figs. 52 and 58).

MS, Page 17:

The swollen MOC@PILs were developed as a new type of supramolecular adsorbents, which were efficient in removal of organic pollutants from water and in purification of organic chemicals. Moreover, through the strategy of deswelling, the MOC@PIL adsorbents could be regenerated.

8. The authors used PIL-PF₆⁻ as the control example to claim that the micropollutants were extracted only by cages. The authors should explain why PIL-PF₆⁻ was used instead of the PIL-NO₃⁻ initially used for the composite synthesis.

Reply: Following the suggestion from the reviewer, the reason of using PIL-PF₆⁻, instead of PIL-NO₃⁻, as the control adsorbent has been added into the revised MS. This is actually resulting from the too large value of the swelling capacity of PIL-NO₃⁻ (Q = 350), as no water sample remained for analysis after swelling, which is also not practical.

MS, Page 15:

Note that pure PIL-PF₆⁻ (Q = 12) was unable to get rid of these pollutants from water, suggesting the important role of the immobilized cage. PIL-PF₆⁻, instead of PIL-NO₃⁻, was

used as the control adsorbent due to the appropriate swellability of the former: the swellability of PIL-NO₃⁻ (Q = 350) was too high and no supernatant water remained for analysis after in-situ swelling of the adsorbent.

REVIEWERS' COMMENTS

Reviewer #1 (Remarks to the Author):

The authors have performed a very comprehensive edit of the manuscript, and addressed all the questions I posed. The citations and literature discussion is much better, the descriptions of molecular recognition and errors are much more consistent, and the technical aspects have been improved. While I may still have concerns about overall novelty, these are really just a matter of my opinion, and others can easily disagree, so this is fine to publish in Nat Comm now.

Reviewer #2 (Remarks to the Author):

I think in general the authors have responded well to the queries of the referees, and I'm happy with the response to all of my original comments.

I do have one further comment. The authors say "the novelty [of this work] resides in how to maintain the flexibility and dynamic character of the cage in the solid state to have solution-level guest binding performance". It is true that they have demonstrated similar binding behaviour within their hybrid materials to the solution-phase, with kinetics and thermodynamics depending on the degree of swelling. However, they have not evidenced their claim that guest uptake in solid-state MOC materials is less appealing or hindered in some way due to reduced flexibility and dynamic behaviour. Can they expand on this point to demonstrate the advantages of their MOC@PILs over solid state materials?

Reviewer #3 (Remarks to the Author):

The revised manuscript looks far better than the original manuscript. I highly appreciate the efforts of the authors for this revision. Particularly, the new section of "Swellability, mechanical properties, and morphology of MOC@PILs" is great and clearly addresses the properties of the resulting gels with different loading amounts. Since many modifications and additional experiments and discussions were added, this version should be recognized as a new manuscript. Though this version is certainly better, I still need to mention several concerns in this manuscript as follows. The authors should address all of these concerns before the acceptance of this manuscript in Nature Communications.

1. The introduction is still redundant. The authors should overhaul the introduction to clearly describe the novelty of this work. I would rather suggest replacing Figure 1 with the conceptual figure of this

work. As another reviewer mentioned, the current Figure 1 does not reflect the work but rather shows the classical concept of host-guest chemistry of coordination cages. This figure should be removed or moved to the SI.

2. As the authors rewrote, the MOC used here is used as an anionic crosslinker for cationic polymers. From this viewpoint, the following sentence should be modified.

“Moreover, the strategy to prepare MOC-integrated gels is currently limited to the use of MOCs as junctions.”

This is because the authors used the MOC as the junction as well.

3. Figure 3g should be replaced with the one with the same magnification as the other SEM images in Figure 3 d-f. The higher magnification image can be used as an inset figure.

Point by point response to the reviewers:

Referee: 1

The authors have performed a very comprehensive edit of the manuscript, and addressed all the questions I posed. The citations and literature discussion is much better, the descriptions of molecular recognition and errors are much more consistent, and the technical aspects have been improved. While I may still have concerns about overall novelty, these are really just a matter of my opinion, and others can easily disagree, so this is fine to publish in Nat Comm now.

Response: We greatly appreciate the positive comments from the reviewer as well as the support of publishing this manuscript in *Nat. Commun.*

Referee: 2

I think in general the authors have responded well to the queries of the referees, and I'm happy with the response to all of my original comments.

Response: We are glad to learn that the reviewer think we have responded well to the queries of the reviewers and we appreciate again the helpful suggestions from reviewer 2 in the last revision.

1. I do have one further comment. The authors say “the novelty [of this work] resides in how to maintain the flexibility and dynamic character of the cage in the solid state to have solution-level guest binding performance”. It is true that they have demonstrated similar binding behaviour within their hybrid materials to the solution-phase, with kinetics and thermodynamics depending on the degree of swelling. However, they have not evidenced their claim that guest uptake in solid-state MOC materials is less appealing or hindered in some way due to reduced flexibility and dynamic behaviour. Can they expand on this point to demonstrate the advantages of their MOC@PILs over solid state materials?

Response: We agree with the reviewer that the measurement of guest uptake ability with the solid-state cage for comparison is very important. We have thus performed binding experiments of the solid-state cage by adding cage solids directly into the liquids of pure guest (norbornadiene and 7-oxabicycloheptane). Results showed that no guest was encapsulated into the cavity of the solid-state cage after stirring the heterogeneous mixture overnight, highlighting the advantage of MOC@PILs for guest binding and the importance of the flexibility and dynamic behavior of the soluble cage in water. These new results and experimental details have been added into the revised MS and the revised Supplementary Information.

MS, Page 11:

The necessity of heating to facilitate binding equilibration further indicated that mechanisms of both portal expansion and vertex dissociation (Fig. 1) may be at work on account of the high energy required.^{33,34,40} Moreover, we have also performed binding experiments of the cage in the solid state by adding cage solids directly into the liquids of pure guest (norbornadiene and 7-oxabicycloheptane). Results proved failure of guest uptake into the cage cavities (Supplementary Fig. 39), also highlighting the importance of the flexibility of dissolved cages in water.

Supplementary Information, Page 26:

Among the four new bicyclic guests, norbornadiene and 7-oxabicycloheptane are liquids. Binding experiments of the Fe^{II}₄L₆ cage in the solid state by adding cage solids directly into

the liquids of pure guest were performed. After stirring the heterogeneous mixtures of the solid-state cages with norbornadiene or 7-oxabicycloheptane at 50 °C overnight, centrifugation was conducted to remove the liquids. The obtained solids were dissolved in D₂O and ¹H NMR spectra were measured immediately. As the binding of norbornadiene and 7-oxabicycloheptane with the dissolved cage in D₂O is slow, the immediate measurement of ¹H NMR could eliminate the possibility of binding the guest molecules initially attached on the solid surface in solution. As shown in Supplementary Figure 39, the ¹H NMR spectra of cages isolated from norbornadiene or 7-oxabicycloheptane only present signals of empty cages together with signals of free guests. We infer the poor binding ability of the solid-state cage resulted from the nonporous nature of the cage solid as well as the poor flexibility and dynamicity of the solid-state cage.

Supplementary Figure 39. ¹H NMR (D₂O, 400 MHz, 298 K) spectra of the Fe^{II}₄L₆ cage and the Fe^{II}₄L₆ cage isolated from the liquid of norbornadiene or 7-oxabicycloheptane. Peaks of free guests have been labelled with orange square.

Referee: 3

The revised manuscript looks far better than the original manuscript. I highly appreciate the efforts of the authors for this revision. Particularly, the new section of “Swellability, mechanical properties, and morphology of MOC@PILs” is great and clearly addresses the properties of the resulting gels with different loading amounts. Since many modifications and additional experiments and discussions were added, this version should be recognized as a new manuscript. Though this version is certainly better, I still need to mention several concerns in this manuscript as follows. The authors should address all of these concerns before the acceptance of this manuscript in Nature Communications.

Response: We appreciate the recognition of our initial effort and are happy to know that the referee is satisfactory with our last revision. We have also carefully addressed the useful issues raised this time by the reviewer, as detailed below.

1. The introduction is still redundant. The authors should overhaul the introduction to clearly describe the novelty of this work. I would rather suggest replacing Figure 1 with the conceptual figure of this work. As another reviewer mentioned, the current Figure 1 does not reflect the work but rather shows the classical concept of host-guest chemistry of coordination cages. This figure should be removed or moved to the SI.

Response: Following the suggestion from the reviewer, we have carefully revised the introduction to be more concise. Moreover, we fully agree with the reviewer that the initial Figure 1 is misleading and unsuitable. As suggested, we have replaced this figure with a conceptual figure of the work in the revised MS.

MS, Page 2:

Guest binding is a complex process and usually gives rise to the most thermodynamically stable host-guest complexes. When designing new capsular hosts for specific guests, it is common to analyze the match in shape and size between them, and the 55% packing coefficient rule (i.e. the ideal filling of a host cavity by a guest bound through weak interactions) established by Rebek is useful in predicting molecular binding.²⁷

Reversible formation of assembled hosts from subunits in solution, on the other hand, allows capsules to open dynamically to accommodate guests facily. Most frequently, the binding and release of guest requires rupture of multiple weak assembling forces to create a suitable opening of capsules, as these capsules usually possess large cavities with small windows. As proposed by Raymond,^{33,34} MOCs as receptors could allow guests that are too large to fit through the windows to enter the cavity by expansion of the windows or by rupture of a metal-ligand bond.

MS, Page 3:

We envisioned that the confined solvent within gels could provide ideal microenvironments for the coordination cages to dynamically move and flex, enabling these cages to bind guests efficiently. Similarly to hard solids, capsule containing soft solids can also be recycled in a heterogeneous fashion. However, to our knowledge, the host-guest chemistry of these integrated MOCs is rarely explored.⁵¹

MS, Page 4:

Fig. 1 | Schematic illustration of the concept of this work. The enlarged inset highlights the two possible binding mechanisms, i.e. portal expansion and vertex dissociation, of a metal-organic cage for the guest molecule that is larger than the portal but can fit within the cavity.

2. As the authors rewrote, the MOC used here is used as an anionic crosslinker for cationic polymers. From this viewpoint, the following sentence should be modified. “Moreover, the strategy to prepare MOC-integrated gels is currently limited to the use of MOCs as junctions.” This is because the authors used the MOC as the junction as well.

Response: We thank the reviewer for pointing this out. The difference between these previous publications and our work is that the reported gels integrate MOCs as covalent junctions. We have thus modified the referred sentence.

MS, Page 3:

The strategy to prepare MOC-integrated gels is also limited to the use of MOCs as covalent junctions.

3. Figure 3g should be replaced with the one with the same magnification as the other SEM images in Figure 3d-f. The higher magnification image can be used as an inset figure.

Response: Suggestion has been followed. We have replaced Figure 3g with an image having the same magnification as the other SEM images, and the initial image having a higher magnification is designed as an inset figure.

MS, Page 8:

Fig. 3 | Swelling capacity, mechanical properties, and morphology of swollen MOC@PILs.

REVIEWER COMMENTS

Reviewer #2

(Remarks to the Author)

I've reviewed the changes that the authors have made, and I think all queries have now been answered well. I support the publication of this manuscript in current form.